# Meta-Query-Net: Resolving Purity-Informativeness Dilemma in Open-set Active Learning

**Dongmin Park[1], Yooju Shin[1], Jihwan Bang[2,3], Youngjun Lee[1], Hwanjun Song[2]\*, Jae-Gil Lee[1]\***

[1] KAIST, [2] NAVER AI Lab, [3] NAVER CLOVA

{dongminpark, yooju.shin, youngjun.lee, jaegil}@kaist.ac.kr
{jihwan.bang, hwanjun.song}@navercorp.com

## Abstract

Unlabeled data examples awaiting annotations contain open-set noise inevitably. A few active learning studies have attempted to deal with this open-set noise for sample selection by filtering out the noisy examples. However, because focusing on the purity of examples in a query set leads to overlooking the informativeness of the examples, the best balancing of purity and informativeness remains an important question. In this paper, to solve this *purity-informativeness dilemma* in open-set active learning, we propose a novel *Meta-Query-Net (MQ-Net)* that adaptively finds the best balancing between the two factors. Specifically, by leveraging the multi-round property of active learning, we train MQ-Net using a query set without an additional validation set. Furthermore, a clear dominance relationship between unlabeled examples is effectively captured by MQ-Net through a novel *skyline* regularization. Extensive experiments on multiple open-set active learning scenarios demonstrate that the proposed MQ-Net achieves $20.14\%$ improvement in terms of accuracy, compared with the state-of-the-art methods.

## 1 Introduction

The success of deep learning in many complex tasks highly depends on the availability of massive data with well-annotated labels, which are very costly to obtain in practice [1]. *Active learning (AL)* is one of the popular learning frameworks to reduce the high human-labeling cost, where a small number of maximally-informative examples are selected by a query strategy and labeled by an oracle repeatedly [2]. Numerous query (*i.e.*, sample selection) strategies, mainly categorized into *uncertainty*-based sampling [3, 4, 5] and *diversity*-based sampling [6, 7, 8], have succeeded in effectively reducing the labeling cost while achieving high model performance.

Despite their success, most standard AL approaches rely on a strict assumption that all unlabeled examples should be cleanly collected from a pre-defined domain called *in-distribution (IN)*, even before being labeled [9]. This assumption is *unrealistic* in practice since the unlabeled examples are mostly collected from rather *casual* data curation processes such as web-crawling. Notably, in the Google search engine, the precision of image retrieval is reported to be $82\%$ on average, and it is worsened to $48\%$ for unpopular entities [10, 11]. That is, such collected unlabeled data naturally involves *open-set noise*, which is defined as a set of the examples collected from different domains called *out-of-distribution (OOD)* [12].

In general, standard AL approaches favor the examples either highly uncertain in predictions or highly diverse in representations as a query for labeling. However, the addition of open-set noise makes these two measures fail to identify informative examples; the OOD examples also exhibit high uncertainty and diversity because they share neither class-distinctive features nor other inductive biases with IN examples [14, 15]. As a result, an active learner is confused and likely to query the OOD examples to

---

*Corresponding authors.

36th Conference on Neural Information Processing Systems (NeurIPS 2022).

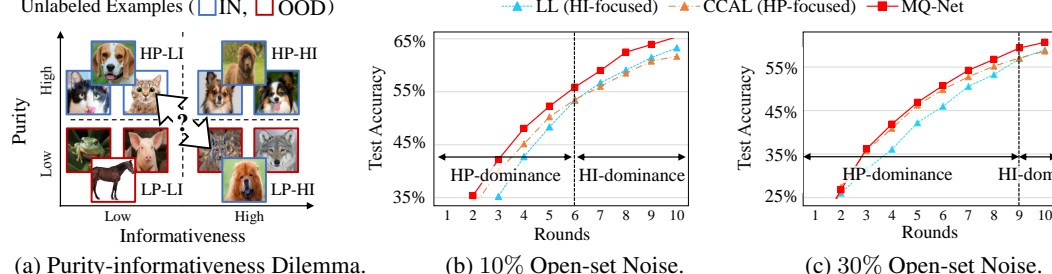

(a) Purity-informativeness Dilemma.     (b) 10% Open-set Noise.     (c) 30% Open-set Noise.

Figure 1: Motivation of MQ-Net: (a) shows the purity-informativeness dilemma for query selection in open-set AL; (b) shows the AL performances of a standard AL method (HI-focused), LL [5], and an open-set AL method (HP-focused), CCAL [13], along with our proposed MQ-Net for the ImageNet dataset with a noise ratio of 10%; (c) shows the trends with a noise ratio of 30%.

a human-annotator for labeling. Human annotators would disregard the OOD examples because they are unnecessary for the target task, thereby wasting the labeling budget. Therefore, the problem of active learning with open-set noise, which we call *open-set active learning*, has emerged as a new important challenge for real-world applications.

Recently, a few studies have attempted to deal with the open-set noise for active learning [13, 16]. They commonly try to increase the purity of examples in a query set, which is defined as the proportion of IN examples, by effectively filtering out the OOD examples. However, whether focusing on the purity is needed *throughout* the entire training period remains a question. In Figure 1(a), let's consider an open-set AL task with a binary classification of cats and dogs, where the images of other animals, *e.g.*, horses and wolves, are regarded as OOD examples. It is clear that the group of high purity and high informativeness (HP-HI) is the most preferable for sample selection. However, when comparing the group of high purity and low informativeness (HP-LI) and that of low purity and high informativeness (LP-HI), the preference between these two groups of examples is *not* clear, but rather contingent on the learning stage and the ratio of OOD examples. Thus, we coin a new term "purity-informativeness dilemma" to call attention to the best balancing of purity and informativeness.

Figures 1(b) and 1(c) illustrate the purity-informativeness dilemma. The standard AL approach, LL[5], puts more weight on the examples of high informativeness (denoted as HI-focused), while the existing open-set AL approach, CCAL [13], puts more weight on those of high purity (denoted as HP-focused). The HP-focused approach improves the test accuracy more significantly than the HI-focused one at earlier AL rounds, meaning that pure as well as easy examples are more beneficial. In contrast, the HI-focused approach beats the HP-focused one at later AL rounds, meaning that highly informative examples should be selected even at the expense of purity. Furthermore, comparing a low OOD (noise) ratio in Figure 1(b) and a high OOD ratio in Figure 1(c), the shift from HP-dominance to HI-dominance tends to occur later at a higher OOD ratio, which renders this dilemma more difficult.

In this paper, to solve the purity-informativeness dilemma in open-set AL, we propose a novel meta-model *Meta-Query-Net (MQ-Net)* that adaptively finds the best balancing between the two factors. A key challenge is the best balancing is unknown in advance. The meta-model is trained to assign higher priority for in-distribution examples over OOD examples as well as for more informative examples among in-distribution ones. The input to the meta-model, which includes the target and OOD labels, is obtained for free from each AL round's query set by leveraging the multi-round property of AL. Moreover, the meta-model is optimized more stably through a novel regularization inspired by the *skyline* query [17, 18] popularly used in multi-objective optimization. As a result, MQ-Net can guide the learning of the target model by providing the best balancing between purity and informativeness throughout the entire training period.

Overall, our main contributions are summarized as follows:

1. We formulate the *purity-informativeness dilemma*, which hinders the usability of open-set AL in real-world applications.
2. As our answer to the dilemma, we propose a novel AL framework, MQ-Net, which keeps finding the best trade-off between purity and informativeness.
3. Extensive experiments on CIFAR10, CIFAR100, and ImageNet show that MQ-Net improves the classifier accuracy consistently when the OOD ratio changes from 10% to 60% by up to 20.14%.

## 2  Related Work

### 2.1  Active Learning and Open-set Recognition

**Active Learning** is a learning framework to reduce the human labeling cost by finding the most informative examples given unlabeled data [9, 19]. One popular direction is uncertainty-based sampling. Typical approaches have exploited prediction probability, *e.g.*, soft-max confidence [20, 3], margin [21], and entropy [22]. Some approaches obtain uncertainty by Monte Carlo Dropout on multiple forward passes [23, 24, 25]. LL [5] predicts the loss of examples by jointly learning a loss prediction module with a target model. Meanwhile, diversity-based sampling has also been widely studied. To incorporate diversity, most methods use a clustering [6, 26] or coreset selection algorithm [7]. Notably, CoreSet [7] finds the set of examples having the highest distance coverage on the entire unlabeled data. BADGE [8] is a hybrid of uncertainty- and diversity-based sampling which uses $k$-means++ clustering in the gradient embedding space. However, this family of approaches is not appropriate for open-set AL since they do not consider how to handle the OOD examples for query selection.

**Open-set Recognition (OSR)** is a detection task to recognize the examples outside of the target domain [12]. Closely related to this purpose, OOD detection has been actively studied [27]. Recent work can be categorized into classifier-dependent, density-based, and self-supervised approaches. The classifier-dependent approach leverages a pre-trained classifier and introduces several scoring functions, such as Uncertainty [28], ODIN [29], mahalanobis distance (MD) [30], and Energy[31]. Recently, ReAct [32] shows that rectifying penultimate activations can enhance most of the aforementioned classifier-dependent OOD scores. The density-based approach learns an auxiliary generative model like a variational auto-encoder to compute likelihood-based OOD scores [33, 34, 35]. Most self-supervised approaches leverage contrastive learning [36, 37, 38]. CSI shows that contrasting with distributionally-shifted augmentations can considerably enhance the OSR performance [36].

The OSR performance of classifier-dependent approaches degrades significantly if the classifier performs poorly [39]. Similarly, the performance of density-based and self-supervised approaches heavily resorts to the amount of clean IN data [35, 36]. Therefore, open-set active learning is a challenging problem to be resolved by simply applying the OSR approaches since it is difficult to obtain high-quality classifiers and sufficient IN data at early AL rounds.

### 2.2  Open-set Active learning

Two recent approaches have attempted to handle the open-set noise for AL [13, 16]. Both approaches try to increase purity in query selection by effectively filtering out the OOD examples. CCAL [13] learns two contrastive coding models each for calculating informativeness and OODness of an example, and combines the two scores using a heuristic balancing rule. SIMILAR [16] selects a pure and core set of examples that maximize the distance on the entire unlabeled data while minimizing the distance to the identified OOD data. However, we found that CCAL and SIMILAR are often worse than standard AL methods, since they always put higher weights on purity although informativeness should be emphasized when the open-set noise ratio is small or in later AL rounds. This calls for developing a new solution to carefully find the best balance between purity and informativeness.

## 3  Purity-Informativeness Dilemma in Open-set Active Learning

### 3.1  Problem Statement

Let $\mathcal{D}_{IN}$ and $\mathcal{D}_{OOD}$ be the IN and OOD data distributions, where the label of examples from $\mathcal{D}_{OOD}$ does not belong to any of the $k$ known labels $Y = \{y_i\}_{i=1}^k$. Then, an unlabeled set is a mixture of IN and OOD examples, $U = \{X_{IN}, X_{OOD}\}$, *i.e.*, $X_{IN} \sim \mathcal{D}_{IN}$ and $X_{OOD} \sim \mathcal{D}_{OOD}$. In the open-set AL, a human oracle is requested to assign a known label $y$ to an IN example $x \in X_{IN}$ with a labeling cost $c_{IN}$, while an OOD example $x \in X_{OOD}$ is marked as open-set noise with a labeling cost $c_{OOD}$.

AL imposes restrictions on the labeling budget $b$ every round. It starts with a small labeled set $S_L$, consisting of both labeled IN and OOD examples. The initial labeled set $S_L$ improves by adding a small but maximally-informative labeled query set $S_Q$ per round, *i.e.*, $S_L \leftarrow S_L \cup S_Q$, where the labeling cost for $S_Q$ by the oracle does not exceed the labeling budget $b$. Hence, the goal of open-set AL is defined to construct the optimal query set $S_Q^*$, minimizing the loss for the *unseen* target IN data. The difference from standard AL is that the labeling cost for OOD examples is introduced, where the labeling budget is wasted when OOD examples are misclassified as informative ones.

Formally, let $C(\cdot)$ be the labeling cost function for a given unlabeled set; then, each round of open-set AL is formulated to find the best query set $S_Q^*$ as

$$S_Q^* = \operatorname*{argmin}_{S_Q:\ C(S_Q) \leq b} \mathbb{E}_{(x,y) \in T_{IN}} \Big[ \ell_{cls}\big(f(x; \Theta_{S_L \cup S_Q}), y\big) \Big],$$

$$\text{where } C(S_Q) = \sum_{x \in S_Q} \big( \mathbb{1}_{[x \in X_{IN}]} c_{IN} + \mathbb{1}_{[x \in X_{OOD}]} c_{OOD} \big). \tag{1}$$

Here, $f(\cdot; \Theta_{S_L \cup S_Q})$ denotes the target model trained on only IN examples in $S_L \cup S_Q$, and $\ell_{cls}$ is a certain loss function, *e.g.*, cross-entropy, for classification. For each AL round, all the examples in $S_Q^*$ are removed from the unlabeled set $U$ and then added to the accumulated labeled set $S_L$ with their labels. This procedure repeats for the total number $r$ of rounds.

## 3.2 Purity-Informativeness Dilemma

An ideal approach for open-set AL would be to increase both purity and informativeness of a query set by completely suppressing the selection of OOD examples and accurately querying the most informative examples among the remaining IN examples. However, the ideal approach is infeasible because overly emphasizing purity in query selection does not promote example informativeness and *vice versa*. Specifically, OOD examples with low purity scores mostly exhibit high informativeness scores because they share neither class-distinctive features nor other inductive biases with the IN examples [14, 15]. We call this trade-off in query selection as the *purity-informativeness dilemma*, which is our new finding expected to trigger a lot of subsequent work.

To address this dilemma, we need to consider the proper weights of a purity score and an informative score when they are combined. Let $\mathcal{P}(x)$ be a purity score of an example $x$ which can be measured by any existing OOD scores, *e.g.*, negative energy [31], and $\mathcal{I}(x)$ be an informativeness score of an example $x$ from any standard AL strategies, *e.g.*, uncertainty [3] and diversity [26]. Next, supposing $z_x = \langle \mathcal{P}(x), \mathcal{I}(x) \rangle$ is a tuple of available purity and informativeness scores for an example $x$. Then, a score combination function $\Phi(z_x)$, where $z_x = \langle \mathcal{P}(x), \mathcal{I}(x) \rangle$, is defined to return an overall score that indicates the necessity of $x$ being included in the query set.

Given two unlabeled examples $x_i$ and $x_j$, if $\mathcal{P}(x_i) > \mathcal{P}(x_j)$ and $\mathcal{I}(x_i) > \mathcal{I}(x_j)$, it is clear to favor $x_i$ over $x_j$ based on $\Phi(z_{x_i}) > \Phi(z_{x_j})$. However, due to the purity-informativeness dilemma, if $\mathcal{P}(x_i) > \mathcal{P}(x_j)$ and $\mathcal{I}(x_i) < \mathcal{I}(x_j)$ or $\mathcal{P}(x_i) < \mathcal{P}(x_j)$ and $\mathcal{I}(x_i) > \mathcal{I}(x_j)$, it is very challenging to determine the dominance between $\Phi(z_{x_i})$ and $\Phi(z_{x_j})$. In order to design $\Phi(\cdot)$, we mainly focus on leveraging *meta-learning*, which is a more agnostic approach to resolve the dilemma other than several heuristic approaches, such as linear combination and multiplication.

# 4 Meta-Query-Net

We propose a meta-model, named *Meta-Query-Net (MQ-Net)*, which aims to learn a meta-score function for the purpose of identifying a query set. In the presence of open-set noise, MQ-Net outputs the meta-score for unlabeled examples to achieve the best balance between purity and informativeness in the selected query set. In this section, we introduce the notion of a self-validation set to guide the meta-model in a supervised manner and then demonstrate the meta-objective of MQ-Net for training. Then, we propose a novel skyline constraint used in optimization, which helps MQ-Net capture the obvious preference among unlabeled examples when a clear dominance exists. Next, we present a way of converting the purity and informativeness scores estimated by existing methods for use in MQ-Net. Note that training MQ-Net is *not* expensive because it builds a light meta-model on a small self-validation set. The overview of MQ-Net is illustrated in Figure 2.

## 4.1 Training Objective with Self-validation Set

The parameters $\mathbf{w}$ contained in MQ-Net $\Phi(\cdot; \mathbf{w})$ is optimized in a supervised manner. For clean supervision, validation data is required for training. Without assuming a hard-to-obtain clean validation set, we propose to use a *self-validation* set, which is instantaneously generated in every AL round. In detail, we obtain a labeled query set $S_Q$ by the oracle, consisting of a labeled IN set and an identified OOD set in every round. Since the query set $S_Q$ is unseen for the target model $\Theta$ and the meta-model $\mathbf{w}$ at the current round, we can exploit it as a self-validation set to train MQ-Net. This self-validation set eliminates the need for a clean validation set in meta-learning.

Given the ground-truth labels in the self-validation set, it is feasible to guide MQ-Net to be trained to resolve the purity-informativeness dilemma by designing an appropriate meta-objective. It is based on the cross-entropy loss for classification because the loss value of training examples has been proven to be effective in identifying high informativeness examples [5]. The conventional loss value by a target model $\Theta$ is masked to be *zero* if $x \in X_{OOD}$ since OOD examples are useless for AL,

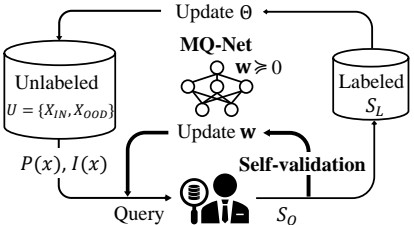

Figure 2: Overview of MQ-Net.

$$\ell_{mce}(x) = \mathbb{1}_{[l_x=1]}\ell_{ce}\big(f(x;\Theta), y\big), \tag{2}$$

where $l$ is a true binary IN label, *i.e.*, 1 for IN examples and 0 for OOD examples, which can be reliably obtained from the self-validation set. This *masked* loss, $\ell_{mce}$, preserves the informativeness of IN examples while excluding OOD examples. Given a self-validation data $S_Q$, the meta-objective is defined such that MQ-Net parameterized by $\mathbf{w}$ outputs a high (or low) meta-score $\Phi(z_x; \mathbf{w})$ if an example $x$'s masked loss value is large (or small),

$$\mathcal{L}(S_Q) = \sum_{i \in S_Q} \sum_{j \in S_Q} \max\Big(0, -\text{Sign}\big(\ell_{mce}(x_i), \ell_{mce}(x_j)\big) \cdot \big(\Phi(z_{x_i}; \mathbf{w}) - \Phi(z_{x_j}; \mathbf{w}) + \eta\big)\Big) \tag{3}$$

$$s.t. \ \forall x_i, x_j, \ \text{if} \ \mathcal{P}(x_i) > \mathcal{P}(x_j) \ \text{and} \ \mathcal{I}(x_i) > \mathcal{I}(x_j), \ \text{then} \ \Phi(z_{x_i}; \mathbf{w}) > \Phi(z_{x_j}; \mathbf{w}),$$

where $\eta > 0$ is a constant margin for the ranking loss, and $\text{Sign}(a, b)$ is an indicator function that returns $+1$ if $a > b$, 0 if $a = b$, and $-1$ otherwise. Hence, $\Phi(z_{x_i}; \mathbf{w})$ is forced to be higher than $\Phi(z_{x_j}; \mathbf{w})$ if $\ell_{mce}(x_i) > \ell_{mce}(x_j)$; in contrast, $\Phi(z_{x_i}; \mathbf{w})$ is forced to be lower than $\Phi(z_{x_j}; \mathbf{w})$ if $\ell_{mce}(x_i) < \ell_{mce}(x_j)$. Two OOD examples do not affect the optimization because they do not have any priority between them, *i.e.*, $\ell_{mce}(x_i) = \ell_{mce}(x_j)$.

In addition to the ranking loss, we add a regularization term named the *skyline* constraint (*i.e.*, the second line) in the meta-objective Eq. (3), which is inspired by the skyline query which aims to narrow down a search space in a large-scale database by keeping only those items that are not worse than any other [17, 18]. Specifically, in the case of $\mathcal{P}(x_i) > \mathcal{P}(x_j)$ and $\mathcal{I}(x_i) > \mathcal{I}(x_j)$, the condition $\Phi(z_{x_i}; \mathbf{w}) > \Phi(z_{x_j}; \mathbf{w})$ must hold in our objective, and hence we make this proposition as the skyline constraint. This simple yet intuitive regularization is very helpful for achieving a meta-model that better judges the importance of purity or informativeness. We provide an ablation study on the skyline constraint in Section 5.4.

## 4.2 Architecture of MQ-Net

MQ-Net is parameterized by a multi-layer perceptron (MLP), a widely-used deep learning architecture for meta-learning [40]. A challenge here is that the proposed skyline constraint in Eq. (3) does not hold with a standard MLP model. To satisfy the skyline constraint, the meta-score function $\Phi(\cdot; \mathbf{w})$ should be a monotonic non-decreasing function because the output (meta-score) of MQ-Net for an example $x_i$ must be higher than that for another example $x_j$ if the two factors (purity and informativeness) of $x_i$ are both higher than those of $x_j$. The MLP model consists of multiple matrix multiplications with non-linear activation functions such as ReLU and Sigmoid. In order for the MLP model to be monotonically non-decreasing, all the parameters in $\mathbf{w}$ for $\Phi(\cdot; \mathbf{w})$ should be *non-negative*, as proven by Theorem 4.1.

**Theorem 4.1.** *For any MLP meta-model $\mathbf{w}$ with non-decreasing activation functions, a meta-score function $\Phi(z; \mathbf{w}) : \mathbb{R}^d \to \mathbb{R}$ holds the skyline constraints if $\mathbf{w} \succeq 0$ and $z(\in \mathbb{R}^d) \succeq 0$, where $\succeq$ is the component-wise inequality.*

*Proof.* An MLP model is involved with matrix multiplication and composition with activation functions, which are characterized by three basic operators: *(1) addition*: $h(z) = f(z) + g(z)$, *(2) multiplication*: $h(z) = f(z) \times g(z)$, and *(3) composition*: $h(z) = f \circ g(z)$. These three operators are guaranteed to be non-decreasing functions if the parameters of the MLP model are all non-negative, because the non-negative weights guarantee all decomposed scalar operations in MLP to be non-decreasing functions. Combining the three operators, the MLP model $\Phi(z; \mathbf{w})$, where $\mathbf{w} \succeq 0$, naturally becomes a monotonic non-decreasing function for each input dimension. Refer to Appendix A for the complete proof. $\square$

In implementation, non-negative weights are guaranteed by applying a ReLU function to meta-model parameters. Since the ReLU function is differentiable, MQ-Net can be trained with the proposed objective in an end-to-end manner. Putting this simple modification, the skyline constraint is preserved successfully without introducing any complex loss-based regularization term. The only remaining condition is that each input of MQ-Net must be a vector of non-negative entries.

### 4.3 Active Learning with MQ-Net

#### 4.3.1 Meta-input Conversion

MQ-Net receives $z_x = \langle \mathcal{P}(x), \mathcal{I}(x) \rangle$ and then returns a meta-score for query selection. All the scores for the input of MQ-Net should be positive to preserve the skyline constraint, *i.e.*, $z \succeq 0$. Existing OOD and AL query scores are converted to the meta-input. The methods used for calculating the scores are orthogonal to our framework. The OOD score $\mathcal{O}(\cdot)$ is conceptually the opposite of purity and varies in its scale; hence, we convert it to a purity score by $\mathcal{P}(x) = \text{Exp}(\text{Normalize}(-\mathcal{O}(x)))$, where $\text{Normalize}(\cdot)$ is the z-score normalization. This conversion guarantees the purity score to be positive. Similarly, for the informativeness score, we convert an existing AL query score $\mathcal{Q}(\cdot)$ to $\mathcal{I}(x) = \text{Exp}(\text{Normalize}(\mathcal{Q}(x)))$. For the z-score normalization, we compute the mean and standard deviation of $\mathcal{O}(x)$ or $\mathcal{Q}(x)$ over the unlabeled examples. Such mean and standard deviation are iteratively computed before the meta-training, and used for the z-score normalization at that round.

#### 4.3.2 Overall Procedure

For each AL round, a target model is trained via stochastic gradient descent (SGD) on mini-batches sampled from the IN examples in the current labeled set $S_L$. Based on the current target model, the purity and informative scores are computed by using certain OOD and AL query scores. The querying phase is then performed by selecting the examples $S_Q$ with the highest meta-scores within the labeling budget $b$. The query set $S_Q$ is used as the self-validation set for training MQ-Net at the current AL round. The trained MQ-Net is used at the next AL round. The alternating procedure of updating the target model and the meta-model repeats for a given number $r$ of AL rounds. The pseudocode of MQ-Net can be found in Appendix B.

## 5 Experiments

### 5.1 Experiment Setting

**Datasets.** We perform the active learning task on three benchmark datasets; CIFAR10 [41], CIFAR100 [41], and ImageNet [42]. Following the 'split-dataset' setup in open-world learning literature [13, 16, 43], we divide each dataset into two subsets: (1) the target set with IN classes and (2) the noise set with OOD classes. Specifically, CIFAR10 is split into the target set with four classes and the noise set with the rest six classes; CIFAR100 into the two sets with 40 and 60 classes; and ImageNet into the two sets with 50 and 950 classes. The entire target set is used as the unlabeled IN data, while only a part of classes in the noise set is selected as the unlabeled OOD data according to the given noise ratio. In addition, following OOD detection literature [28, 33], we also consider the 'cross-dataset' setup, which mixes a certain dataset with two external OOD datasets collected from different domains, such as LSUN [44] and Places365 [45]. For sake of space, we present all the results on the cross-dataset setup in Appendix D.

**Algorithms.** We compare MQ-Net with a random selection, four standard AL, and two recent open-set AL approaches.

- *Standard AL*: The four methods perform AL without any processing for open-set noise: (1) CONF [3] queries the most uncertain examples with the lowest softmax confidence in the prediction, (2) CORESET [7] queries the most diverse examples with the highest coverage in the representation space, (3) LL [5] queries the examples having the largest predicted loss by jointly learning a loss prediction module, and (4) BADGE [8] considers both uncertainty and diversity by querying the most representative examples in the gradient via $k$-means++ clustering [46].

- *Open-set AL*: The two methods tend to put more weight on the examples with high purity: (1) CCAL [13] learns two contrastive coding models for calculating informativeness and OODness, and then it combines the two scores into one using a heuristic balancing rule, and (2) SIMILAR [16] selects a pure and core set of examples that maximize the distance coverage on the entire unlabeled data while minimizing the distance coverage to the already labeled OOD data.

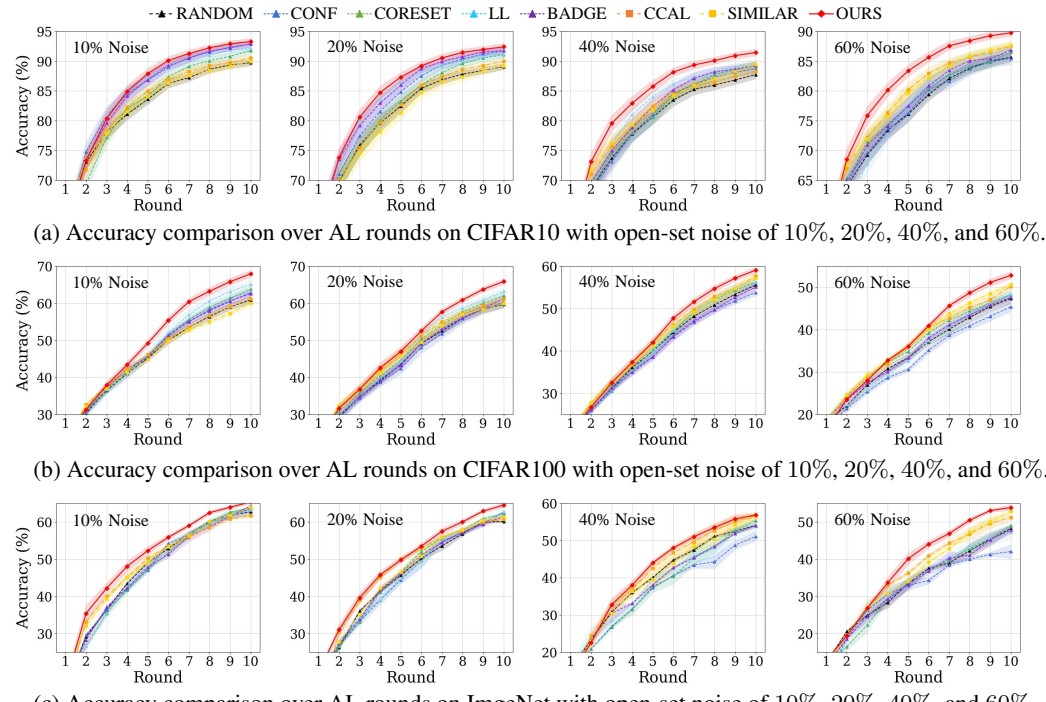

(a) Accuracy comparison over AL rounds on CIFAR10 with open-set noise of 10%, 20%, 40%, and 60%.

(b) Accuracy comparison over AL rounds on CIFAR100 with open-set noise of 10%, 20%, 40%, and 60%.

(c) Accuracy comparison over AL rounds on ImgeNet with open-set noise of 10%, 20%, 40%, and 60%.

Figure 3: Test accuracy over AL rounds for CIFAR10, CIFAR100, and ImageNet with varying open-set noise ratios.

For all the experiments, regarding the two inputs of MQ-Net, we mainly use CSI [36] and LL [5] for calculating the purity and informativeness scores, respectively. For CSI, as in CCAL, we train a contrastive learner on the entire unlabeled set with open-set noise since the clean in-distribution set is not available in open-set AL. The ablation study in Section 5.4 shows that MQ-Net is also effective with other OOD and AL scores as its input.

**Implementation Details.** We repeat the three steps—training, querying, and labeling—of AL. The total number $r$ of rounds is set to 10. Following the prior open-set AL setup [13, 16], we set the labeling cost $c_{IN} = 1$ for IN examples and $c_{OOD} = 1$ for OOD examples. For the class-split setup, the labeling budget $b$ per round is set to $500$ for CIFAR10/100 and $1,000$ for ImageNet. Regarding the open-set noise ratio $\tau$, we configure four different levels from light to heavy noise in $\{10\%, 20\%, 40\%, 60\%\}$. In the case of $\tau = 0\%$ (no noise), MQ-Net naturally discards the purity score and only uses the informativeness score for query selection, since the self-validation set does not contain any OOD examples. The initial labeled set is randomly selected uniformly at random from the entire unlabeled set within the labeling budget $b$. For the architecture of MQ-Net, we use a 2-layer MLP with the hidden dimension size of 64 and the Sigmoid activation fuction. We report the average results of five runs with different class splits. We did *not* use any pre-trained networks. See Appendix C for more implementation details with training configurations. All methods are implemented with PyTorch 1.8.0 and executed on a single NVIDIA Tesla V100 GPU. The code is available at `https://github.com/kaist-dmlab/MQNet`.

## 5.2 Open-set Noise Robustness

### 5.2.1 Results over AL Rounds

Figure 3 illustrates the test accuracy of the target model over AL rounds on the two CIFAR datasets. MQ-Net achieves the highest test accuracy in most AL rounds, thereby reaching the best test accuracy at the final round in every case for various datasets and noise ratios. Compared with the two existing open-set AL methods, CCAL and SIMILAR, MQ-Net shows a steeper improvement in test accuracy over rounds by resolving the purity-informativeness dilemma in query selection. For example, the performance gap between MQ-Net and the two open-set AL methods gets larger after the sixth round, as shown in Figure 3(b), because CCAL and SIMILAR mainly depend on purity in query selection, which conveys less informative information to the classifier. For a better classifier, informative

Table 1: Last test accuracy (%) at the final round for CIFAR10, CIFAR100, and ImageNet. The best results are in bold, and the second best results are underlined.

| Datasets | | CIFAR10 (4:6 split) | | | | CIFAR100 (40:60 split) | | | | ImageNet (50:950 split) | | | |
|---|---|---|---|---|---|---|---|---|---|---|---|---|---|
| Noise Ratio | | 10% | 20% | 40% | 60% | 10% | 20% | 40% | 60% | 10% | 20% | 40% | 60% |
| Non-AL | RANDOM | 89.83 | 89.06 | 87.73 | 85.64 | 60.88 | 59.69 | 55.52 | 47.37 | 62.72 | 60.12 | 54.04 | 48.24 |
| Standard AL | CONF | 92.83 | 91.72 | 88.69 | 85.43 | 62.84 | 60.20 | 53.74 | 45.38 | 63.56 | 62.56 | 51.08 | 45.04 |
| | CORESET | 91.76 | 91.06 | 89.12 | 86.50 | 63.79 | 62.02 | 56.21 | 48.33 | 63.64 | 62.24 | 55.32 | 49.04 |
| | LL | 92.09 | 91.21 | 89.41 | 86.95 | 65.08 | 64.04 | 56.27 | 48.49 | 63.28 | 61.56 | 55.68 | 47.30 |
| | BADGE | 92.80 | 91.73 | 89.27 | 86.83 | 62.54 | 61.28 | 55.07 | 47.60 | 64.84 | 61.48 | 54.04 | 47.80 |
| Open-set AL | CCAL | 90.55 | 89.99 | 88.87 | 87.49 | 61.20 | 61.16 | 56.70 | 50.20 | 61.68 | 60.70 | 56.60 | 51.16 |
| | SIMILAR | 89.92 | 89.19 | 88.53 | 87.38 | 60.07 | 59.89 | 56.13 | 50.61 | 63.92 | 61.40 | 56.48 | 52.84 |
| Proposed | **MQ-Net** | **93.10** | **92.10** | **91.48** | **89.51** | **66.44** | **64.79** | **58.96** | **52.82** | **65.36** | **63.08** | **56.95** | **54.11** |
| *% improve over 2nd best* | | 0.32 | 0.40 | 2.32 | 2.32 | 2.09 | 1.17 | 3.99 | 4.37 | 0.80 | 1.35 | 0.62 | 2.40 |
| *% improve over the least* | | 3.53 | 3.26 | 3.33 | 4.78 | 10.6 | 8.18 | 9.71 | 16.39 | 5.97 | 3.92 | 11.49 | 20.14 |

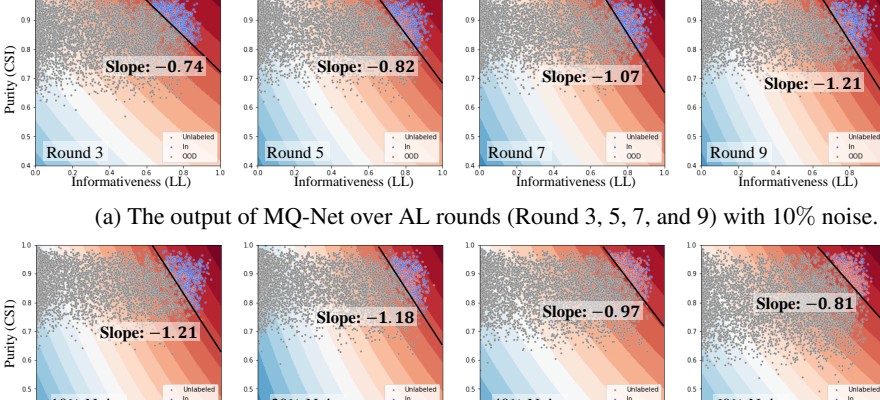

(a) The output of MQ-Net over AL rounds (Round 3, 5, 7, and 9) with 10% noise.

(b) The final round's output of MQ-Net with varying noise ratios (10%, 20%, 40%, and 60%).

Figure 4: Visualization of the query score distribution of MQ-Net on CIFAR100. $x$- and $y$-axis indicate the normalized informativeness and purity scores, respectively. The background color represents the query score of MQ-Net; the red is high, and the blue is low. Gray points represent unlabeled data, and blue and red points are the IN and OOD examples in the query set, respectively. The slope of the tangent line on the lowest scored example in the query set is displayed together; the steeper the slope, the more informativeness is emphasized in query selection.

examples should be favored at a later AL round due to the sufficient number of IN examples in the labeled set. In contrast, MQ-Net keeps improving the test accuracy even in a later AL round by finding the best balancing between purity and informativeness in its query set. More analysis of MQ-Net associated with the purity-informativeness dilemma is discussed in Section 5.3.

### 5.2.2 Results with Varying Noise Ratios

Table 1 summarizes the last test accuracy at the final AL round for three datasets with varying levels of open-set noise. Overall, the last test accuracy of MQ-Net is the best in every case. This superiority concludes that MQ-Net successfully finds the best trade-off between purity and informativeness in terms of AL accuracy regardless of the noise ratio. In general, the performance improvement becomes larger with the increase in the noise ratio. On the other hand, the two open-set AL approaches are even worse than the four standard AL approaches when the noise ratio is less than or equal to 20%. Especially, in CIFAR10 relatively easier than others, CCAL and SIMILAR are inferior to the non-robust AL method, LL, even with 40% noise. This trend confirms that increasing informativeness is more crucial than increasing purity when the noise ratio is small; highly informative examples are still beneficial when the performance of a classifier is saturated in the presence of open-set noise. An in-depth analysis on the low accuracy of the existing open-set AL approaches in a low noise ratio is presented in Appendix E.

Table 2: Effect of the meta inputs on MQ-Net.

| Dataset | | CIFAR10 (4:6 split) | | | |
|---|---|---|---|---|---|
| Noise Ratio | | 10% | 20% | 40% | 60% |
| Standard AL | BADGE | 92.80 | 91.73 | 89.27 | 86.83 |
| Open-set AL | CCAL | 90.55 | 89.99 | 88.87 | 87.49 |
| MQ-Net | CONF-ReAct | 93.21 | 91.89 | 89.54 | 87.99 |
| | CONF-CSI | **93.28** | **92.40** | 91.43 | 89.37 |
| | LL-ReAct | 92.34 | 91.85 | 90.08 | 88.41 |
| | LL-CSI | 93.10 | 92.10 | **91.48** | **89.51** |

Table 3: Efficacy of the self-validation set.

| Dataset | | CIFAR10 (4:6 split) | | | |
|---|---|---|---|---|---|
| Noise Ratio | | 10% | 20% | 40% | 60% |
| MQ-Net | Query set | **93.10** | **92.10** | **91.48** | **89.51** |
| | Random | 92.10 | 91.75 | 90.88 | 87.65 |

Table 4: Efficacy of the skyline constraint.

| Noise Ratio | | 10% | 20% | 40% | 60% |
|---|---|---|---|---|---|
| MQ-Net | w/ skyline | **93.10** | **92.10** | **91.48** | **89.51** |
| | w/o skyline | 87.25 | 86.29 | 83.61 | 81.67 |

## 5.3 Answers to the Purity-Informativeness Dilemma

The high robustness of MQ-Net in Table 1 and Figure 3 is mainly attributed to its ability to keep finding the best trade-off between purity and informativeness. Figure 4(a) illustrates the preference change of MQ-Net between purity and informativeness throughout the AL rounds. As the round progresses, MQ-Net automatically raises the importance of informativeness rather than purity; the slope of the tangent line keeps steepening from $-0.74$ to $-1.21$. This trend implies that more informative examples are required to be labeled when the target classifier becomes mature. That is, as the model performance increases, 'fewer but highly-informative' examples are more impactful than 'more but less-informative' examples in terms of improving the model performance. Figure 4(b) describes the preference change of MQ-Net with varying noise ratios. Contrary to the trend over AL rounds, as the noise ratio gets higher, MQ-Net prefers purity more over informativeness.

## 5.4 Ablation Studies

**Various Combination of Meta-input.** MQ-Net can design its purity and informativeness scores by leveraging diverse metrics in the existing OOD detection and AL literature. Table 2 shows the final round test accuracy on CIFAR10 for the four variants of score combinations, each of which is constructed by a combination of two purity scores and two informativeness scores; each purity score is induced by the two recent OOD detection methods, ReAct [32] and CSI [36], while each informativeness score is converted from the two existing AL methods, CONF and LL. "CONF-ReAct" denotes a variant that uses ReAct as the purity score and CONF as the informativeness score.

Overall, all variants perform better than standard and open-set AL baselines in every noise level. Refer to Table 2 for detailed comparison. This result concludes that MQ-Net can be generalized over different types of meta-input owing to the learning flexibility of MLPs. Interestingly, the variant using CSI as the purity score is consistently better than those using ReAct. ReAct, a classifier-dependent OOD score, performs poorly in earlier AL rounds. A detailed analysis of the two OOD detectors, ReAct and CSI, over AL rounds can be found in Appendix F.

**Efficacy of Self-validation Set.** MQ-Net can be trained with an independent validation set, instead of using the proposed self-validation set. We generate the independent validation set by randomly sampling the same number of examples as the self-validation set with their ground-truth labels from the entire data not overlapped with the unlabeled set used for AL. As can be seen from Table 3, it is of interest to see that our self-validation set performs better than the random validation set. The two validation sets have a major difference in data distributions; the self-validation set mainly consists of the examples with highest meta-scores among the remaining unlabeled data per round, while the random validation set consists of random examples. We conclude that the meta-score of MQ-Net has the potential for constructing a high-quality validation set in addition to query selection.

**Efficacy of Skyline Constraint.** Table 4 demonstrates the final round test accuracy of MQ-Net with or without the skyline constraint. For the latter, a standard 2-layer MLP is used as the meta-network architecture without any modification. The performance of MQ-Net degrades significantly without the skyline constraint, meaning that the non-constrained MLP can easily overfit to the small-sized self-validation set, thereby assigning high output scores on less-pure and less-informative examples. Therefore, the violation of the skyline constraint in optimization makes MQ-Net hard to balance between the purity and informativeness scores in query selection.

**Efficacy of Meta-objective.** MQ-Net keeps finding the best balance between purity and informativeness over multiple AL rounds by repeatedly minimizing the meta-objective in Eq. (3). To validate its

Table 5: Efficacy of the meta-objective in MQ-Net. We show the AL performance of two alternative balancing rules compared with MQ-Net for the split-dataset setup on CIFAR10 with the open-set noise ratios of 20% and 40%.

| Dataset | Noise Ratio | Round | 1 | 2 | 3 | 4 | 5 | 6 | 7 | 8 | 9 | 10 |
|---|---|---|---|---|---|---|---|---|---|---|---|---|
| CIFAR10 (4:6 split) | 20% | $\mathcal{P}(x) + \mathcal{I}(x)$ | **61.93** | **73.82** | 76.16 | 80.65 | 82.61 | 85.73 | 87.44 | 88.86 | 89.21 | 89.49 |
| | | $\mathcal{P}(x) \cdot \mathcal{I}(x)$ | **61.93** | 71.79 | 78.09 | 81.32 | 84.16 | 86.39 | 88.74 | 89.89 | 90.54 | 91.20 |
| | | MQ-Net | **61.93** | **73.82** | **80.58** | **84.72** | **87.31** | **89.20** | **90.52** | **91.46** | **91.93** | **92.10** |
| | 40% | $\mathcal{P}(x) + \mathcal{I}(x)$ | **59.31** | **72.50** | 75.67 | 78.78 | 81.70 | 83.74 | 85.08 | 86.48 | 87.47 | 88.86 |
| | | $\mathcal{P}(x) \cdot \mathcal{I}(x)$ | **59.31** | 66.37 | 73.57 | 77.85 | 81.37 | 84.22 | 86.80 | 88.04 | 88.73 | 89.11 |
| | | MQ-Net | **59.31** | **72.50** | **79.54** | **82.94** | **85.77** | **88.16** | **89.34** | **90.07** | **90.92** | **91.48** |

efficacy, we compare it with two simple alternatives based on heuristic balancing rules such as *linear combination* and *multiplication*, denoted as $\mathcal{P}(x) + \mathcal{I}(x)$ and $\mathcal{P}(x) \cdot \mathcal{I}(x)$, respectively. Following the default setting of MQ-Net, we use LL for $\mathcal{P}(x)$ and CSI for $\mathcal{I}(x)$.

Table 5 shows the AL performance of the two alternatives and MQ-Net for the split-dataset setup on CIFAR10 with the noise ratios of 20% and 40%. MQ-Net beats the two alternatives after the second AL round where MQ-Net starts balancing purity and informativeness with its meta-objective. This result implies that our meta-objective successfully finds the best balance between purity and informativeness by emphasizing informativeness over purity at the later AL rounds.

## 5.5 Effect of Varying OOD Labeling Cost

The labeling cost for OOD examples could vary with respect to data domains. To validate the robustness of MQ-Net on diverse labeling scenarios, we conduct an additional study of adjusting the labeling cost $c_{OOD}$ for the OOD examples. Table 6 summarizes the performance change with four different labeling costs (*i.e.*, 0.5, 1, 2, and 4). The two standard AL methods, CONF and CORESET, and two open-set AL methods, CCAL and SIMILAR, are compared with MQ-Net. Overall, MQ-Net consistently outperforms the four baselines regardless of the labeling cost. Meanwhile, CCAL and SIMILAR are more robust to the higher labeling cost than CONF and CORESET;

Table 6: Effect of varying the labeling cost $c_{OOD}$. We measure the last test accuracy for the split-dataset setup on CIFAR10 with an open-set noise ratio of 40%. The best values are in bold.

| $c_{OOD}$ | 0.5 | 1 | 2 | 4 |
|---|---|---|---|---|
| CONF | 91.05 | 88.69 | 86.25 | 80.06 |
| CORESET | 90.59 | 89.12 | 85.32 | 81.22 |
| CCAL | 90.25 | 88.87 | 88.16 | 87.25 |
| SIMILAR | 91.05 | 88.69 | 87.95 | 86.52 |
| MQ-Net | **92.52** | **91.48** | **89.53** | **87.36** |

CCAL and SIMILAR, which favor high purity examples, query more IN examples than CONF and CORESET, so they are less affected by the labeling cost, especially when it is high.

## 6 Conclusion

We propose MQ-Net, a novel meta-model for open-set active learning that deals with the purity-informativeness dilemma. MQ-Net finds the best balancing between the two factors, being adaptive to the noise ratio and target model status. A clean validation set for the meta-model is obtained for free by exploiting the procedure of active learning. A ranking loss with the skyline constraint optimizes MQ-Net to make the output a legitimate meta-score that keeps the obvious order of two examples. MQ-Net is shown to yield the best test accuracy throughout the entire active learning rounds, thereby empirically proving the correctness of our solution to the purity-informativeness dilemma. Overall, we expect that our work will raise the practical usability of active learning with open-set noise.

## Acknowledgement

This work was supported by Institute of Information & Communications Technology Planning & Evaluation (IITP) grant funded by the Korea government (MSIT) (No. 2020-0-00862, DB4DL: High-Usability and Performance In-Memory Distributed DBMS for Deep Learning). The experiment was conducted by the courtesy of NAVER Smart Machine Learning (NSML) [47].

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
