# Meta-Query-Net: Resolving Purity-Informativeness Dilemma in Open-set Active Learning
## (Supplementary Material)

## A  Complete Proof of Theorem 4.1

Let $z_x = \{z_x^{\langle 1 \rangle}, \ldots, z_x^{\langle d \rangle}\}$ be the $d$-dimensional meta-input for an example $x$ consisting of $d$ available purity and informativeness scores.[2] A non-negative-weighted MLP $\Phi_{\mathbf{w}}$ can be formulated as

$$h^{[l]} = \sigma\big(W^{[l]} \cdot h^{[l-1]} + b^{[l]}\big), \ l \in \{1, \ldots, L\}, \tag{4}$$

where $h^{[0]} = z_x, h^{[L]} \in \mathbb{R}, W^{[l]} \succeq 0$, and $b^{[l]} \succeq 0$; $L$ is the number of layers and $\sigma$ is a non-linear activation function.

We prove Theorem 4.1 by mathematical induction, as follows: (1) the first layer's output satisfies the skyline constraint by Lemmas A.1 and A.2; and (2) the $k$-th layer's output ($k \geq 2$) also satisfies the skyline constraint if the $(k-1)$-th layer's output satisfies the skyline constraint. Therefore, we conclude that the skyline constraint holds for any non-negative-weighted MLP $\Phi(z; \mathbf{w})\colon \mathbb{R}^d \to \mathbb{R}$ by Theorem A.4.

**Lemma A.1.** *Let $g^{[1]}(z_x) = W^{[1]} \cdot z_x + b^{[1]}$ be a non-negative-weighted single-layer MLP with $m$ hidden units and an identity activation function, where $W^{[1]} \in \mathbb{R}^{m \times d} \succeq 0$ and $b^{[1]} \in \mathbb{R}^m \succeq 0$. Given the meta-input of two different examples $z_{x_i}$ and $z_{x_j}$, the function $g^{[1]}(z_x)$ satisfies the skyline constraint as*

$$z_{x_i} \succeq z_{x_j} \implies g^{[1]}(z_{x_i}) \succeq g^{[1]}(z_{x_j}). \tag{5}$$

*Proof.* Let $g^{[1]}(z_x)$ be $g(z_x)$ and $W^{[1]}$ be $W$ for notation simplicity. Consider each dimension's scalar output of $g(z_x)$, and it is denoted as $g^{\langle p \rangle}(z_x)$ where $p$ is an index of the output dimension. Similarly, let $W^{\langle p,n \rangle}$ be a scalar element of the matrix $W$ on the $p$-th row and $n$-th column. With the matrix multiplication, the scalar output $g^{\langle p \rangle}(z_x)$ can be considered as the sum of multiple scalar linear operators $W^{\langle p,n \rangle} \cdot z_x^{\langle n \rangle}$. By this property, we show that $g^{\langle p \rangle}(z_{x_i}) - g^{\langle p \rangle}(z_{x_j}) \geq 0$ if $z_{x_i} \succeq z_{x_j}$ by

$$g^{\langle p \rangle}(z_{x_i}) - g^{\langle p \rangle}(z_{x_j}) = W^{\langle p,\cdot \rangle} \cdot z_{x_i} - W^{\langle p,\cdot \rangle} \cdot z_{x_j} = \sum_{n=1}^{d} \big(W^{\langle p,n \rangle} \cdot z_{x_i}^{\langle n \rangle} - W^{\langle p,n \rangle} \cdot z_{x_j}^{\langle n \rangle}\big)$$
$$= \sum_{n=1}^{d} \big(W^{\langle p,n \rangle} \cdot (z_{x_i}^{\langle n \rangle} - z_{x_j}^{\langle n \rangle})\big) \geq 0. \tag{6}$$

Therefore, without loss of generality, $g(z_{x_i}) - g(z_{x_j}) \succeq 0$ if $z_{x_i} \succeq z_{x_j}$. This concludes the proof. □

**Lemma A.2.** *Let $h(z_x) = \sigma(g(z_x))$ where $\sigma$ is a non-decreasing non-linear activation function. If the skyline constraint holds by $g(\cdot) \in \mathbb{R}^d$, the function $h(z_x)$ also satisfies the skyline constraint as*

$$z_{x_i} \succeq z_{x_j} \implies h(z_{x_i}) \succeq h(z_{x_j}). \tag{7}$$

*Proof.* By the composition rule of the non-decreasing function, applying any non-decreasing function does not change the order of its inputs. Therefore, $\sigma(g(z_{x_i})) - \sigma(g(z_{x_j})) \succeq 0$ if $g(z_{x_i}) \succeq g(z_{x_j})$. □

**Lemma A.3.** *Let $h^{[k]}(z_x) = \sigma\big(W^{[k]} \cdot h^{[k-1]}(z_x) + b^{[k]}\big)$ be the $k$-th layer of a non-negative-weighted MLP ($k \geq 2$), where $W^{[k]} \in \mathbb{R}^{m' \times m} \succeq 0$ and $b^{[k]} \in \mathbb{R}^{m'} \succeq 0$. If $h^{[k-1]}(\cdot) \in \mathbb{R}^m$ satisfies the skyline constraint, the function $h^{[k]}(z_x)$ also holds the skyline constraint as*

$$z_{x_i} \succeq z_{x_j} \implies h^{[k]}(z_{x_i}) \succeq h^{[k]}(z_{x_j}). \tag{8}$$

---

[2] We use only two scores ($d = 2$) in MQ-Net, one for purity and another for informativeness.

*Proof.* Let $W^{[k]}$ be $W$, $h^{[k]}(z_x)$ be $h(z_x)$, and $h^{[k-1]}(z_x)$ be $h_{input}(z_x)$ for notation simplicity. Since an intermediate layer uses $h_{input}(z_x)$ as its input rather than $z$, Eq. (6) changes to

$$g^{\langle p \rangle}(z_{x_i}) - g^{\langle p \rangle}(z_{x_j}) = \sum_{n=1}^{d} \left( W^{\langle p,n \rangle} \cdot \left( h_{input}^{\langle n \rangle}(z_{x_i}) - h_{input}^{\langle n \rangle}(z_{x_j}) \right) \right) \geq 0, \qquad (9)$$

where $g^{\langle p \rangle}(z_{x_i})$ is the $p$-th dimension's output before applying non-linear activation $\sigma$. Since $h_{input}(\cdot)$ satisfies the skyline constraint, $h_{input}^{\langle n \rangle}(z_{x_i}) > h_{input}^{\langle n \rangle}(z_{x_j})$ when $z_{x_i} \succeq z_{x_j}$, $g^{\langle p \rangle}(z_{x_i}) > g^{\langle p \rangle}(z_{x_j})$ for all $p \in \{1, \dots, m'\}$. By Lemma A.2, $h^{\langle p \rangle}(z_{x_i}) - h^{\langle p \rangle}(z_{x_j}) = \sigma(g^{\langle p \rangle}(z_{x_i})) - \sigma(g^{\langle p \rangle}(z_{x_j})) \geq 0$ for all $p$. Therefore, $z_{x_i} \succeq z_{x_j} \implies h^{[k]}(z_{x_i}) \succeq h^{[k]}(z_{x_j})$. $\qquad \square$

**Theorem A.4.** *For any non-negative-weighted MLP $\Phi(z; \boldsymbol{w}) \colon \mathbb{R}^d \to \mathbb{R}$ where $\boldsymbol{w} \succeq 0$, the skyline constraint holds such that $z_{x_i} \succeq z_{x_j} \implies \Phi(z_{x_i}) \geq \Phi(z_{x_j}) \, \forall z_{x_i}, z_{x_j} \in \mathbb{R}^d \succeq 0$.*

*Proof.* By mathematical induction, where Lemmas A.1 and A.2 constitute the base step, and Lemma A.3 is the inductive step, any non-negative-weighted MLP satisfies the skyline constraint. $\quad \square$

# B    Detailed Procedure of MQ-Net

**Mini-batch Optimization.** Mini-batch examples are sampled from the labeled query set $S_Q$ which contains both IN and OOD examples. Since the meta-objective in Eq. (3) is a ranking loss, a mini-batch $\mathcal{M}$ is a set of meta-input pairs such that $\mathcal{M} = \{(z_{x_i}, z_{x_j}) | \; x_i, x_j \in S_Q\}$ where $z_x = \langle \mathcal{P}(x), \mathcal{I}(x) \rangle$. To construct a paired mini-batch $\mathcal{M}$ of size $M$, we randomly sample $2M$ examples from $S_Q$ and pair the $i$-th example with the $(M+i)$-th one for all $i \in \{1, \dots, M\}$. Then, the loss for mini-batch optimization of MQ-Net is defined as

$$\mathcal{L}_{meta}(\mathcal{M}) = \sum_{(i,j) \in \mathcal{M}} \max\Big(0, -\text{Sign}\big(\ell_{mce}(x_i), \ell_{mce}(x_j)\big) \cdot \big(\Phi(z_{x_i}; \mathbf{w}) - \Phi(z_{x_j}; \mathbf{w}) + \eta\big)\Big) : \mathbf{w} \succeq 0. \quad (10)$$

**Algorithm Pseudocode.** Algorithm 1 describes the overall active learning procedure with MQ-Net, which is self-explanatory. For each AL round, a target model $\Theta$ is trained via stochastic gradient descent (SGD) using IN examples in the labeled set $S_L$ (Lines 3–5). This trained target model is saved as the final target model at the current round. Next, the querying phase is performed according

---

**Algorithm 1** AL Procedure with MQ-Net

INPUT:  $S_L$: labeled set, $U$: unlabeled set, $r$: number of rounds, $b$: labeling budget, $C$: cost function, $\Theta$: parameters of the target model, $\mathbf{w}$: parameters of MQ-Net
OUTPUT:  Final target model $\Theta_*$

1:  $\mathbf{w} \leftarrow$ Initialize the meta-model parameters;
2:  **for** $r = 1$ **to** $r$ **do**
3:      /* Training the target model parameterized by $\Theta$*/
4:      $\Theta \leftarrow$ Initialize the target model parameters;
5:      $\Theta \leftarrow \text{TrainingClassifier}(S_L, \Theta)$;
6:      /* Querying for the budget $b$ */
7:      $S_Q \leftarrow \emptyset$;
8:      **while** $C(S_Q) \leq b$ **do**
9:          **if** $r = 1$ **do**
10:             $S_Q \leftarrow S_Q \cup \arg\max_{x \in U}(\mathcal{P}(x) + \mathcal{I}(x))$;
11:         **else do**
12:             $S_Q \leftarrow S_Q \cup \arg\max_{x \in U}(\Phi(x; \mathbf{w}))$;
13:     $S_L \leftarrow S_L \cup S_Q$;  $U \leftarrow U \backslash S_Q$
14:     /* Training MQ-Net $\Phi$ parameterized by $\mathbf{w}$ */
15:     **for** $t = 1$ **to** meta-train-steps **do**
16:         Draw a mini-batch $\mathcal{M}$ and from $S_Q$;
17:         $\mathbf{w} \leftarrow \mathbf{w} - \alpha \nabla_{\mathbf{w}}\big(\mathcal{L}_{meta}(\mathcal{M})\big)$;
18: **return** $\Theta$;

---

to the order of meta-query scores from $\Phi$ given the budget $b$ (Lines 6–13). Then, the meta-training phase is performed, and the meta-model $\mathbf{w}$ is updated via SGD using the labeled query set $S_Q$ as a self-validation set (Lines 14–17). Lines 3–17 repeat for the given number $r$ of rounds. At the first round, because there is no meta-model trained in the previous round, the query set is constructed by choosing the examples whose sum of purity and informativeness scores is the largest (Lines 9–10).

## C    Implementation Details

### C.1    Split-dataset Setup

**Training Configurations.** We train ResNet-18 using SGD with a momentum of 0.9 and a weight decay of 0.0005, and a batch size of 64. The initial learning rate of 0.1 is decayed by a factor of 0.1 at 50% and 75% of the total training iterations. In the setup of open-set AL, the number of IN examples for training differs depending on the query strategy. We hence use a fixed number of training iterations instead of epochs for fair optimization. The number of training iterations is set to 20,000 for CIFAR10/100 and 30,000 for ImageNet. We set $\eta$ to 0.1 for all cases. We train MQ-Net for 100 epochs using SGD with a weight decay of 0.0005, and a mini-batch size of 64. An initial learning rate of 0.01 is decayed by a factor of 0.1 at 50% of the total training iterations. Since MQ-Net is not trained at the querying phase of the first AL round, we simply use the linear combination of purity and informativeness as the query score, *i.e.*, $\Phi(x) = \mathcal{P}(x) + \mathcal{I}(x)$. For calculating the CSI-based purity score, we train a contrastive learner for CSI with 1,000 epochs under the LARS optimizer with a batch size of 32. Following CCAL [48], we use the distance between each unlabeled example to the closest OOD example in the labeled set on the representation space of the contrastive learner as the OOD score. The hyperparameters for other algorithms are favorably configured following the original papers.

### C.2    Cross-dataset Setup

**Datasets.** Each of CIFAR10, CIFAR100, and ImageNet is mixed with OOD examples sampled from an OOD dataset combined from two different domains—LSUN [44], an indoor scene understanding dataset of 59M images with 10 classes, and Places365 [45], a large collection of place scene images with 365 classes. The resolution of LSUN and Places365 is resized into $32 \times 32$ after random cropping when mixing with CIFAR10 and CIFAR100. For ImageNet, as in the split-dataset setup in Section 5.1, we use 50 randomly-selected classes as IN examples, namely ImageNet50.

**Implementation Details.** For the cross-dataset setup, the budget $b$ is set to $1,000$ for CIFAR-10 and ImageNet50 and $2,000$ for CIFAR-100 following the literature [5]. Regarding the open-set noise ratio, we also configure four different levels from light to heavy noise in $\{10\%, 20\%, 40\%, 60\%\}$. The initial labeled set is selected uniformly at random from the entire unlabeled set within the labeling budget $b$. For instance, when $b$ is $1,000$ and $\tau$ is $20\%$, 800 IN examples and 200 OOD examples are expected to be selected as the initial set.

## D    Experiment Results on Cross-datasets

### D.1    Results over AL Rounds

Figure 5 shows the test accuracy of the target model throughout AL rounds on the three cross-datasets. Overall, as analyzed in Section 5.2, MQ-Net achieves the highest test accuracy in most AL rounds, thereby reaching the best test accuracy at the final round in every case of various datasets and noise ratios. Compared with the two existing open-set AL methods, CCAL and SIMILAR, MQ-Net shows a steeper improvement in test accuracy over rounds by resolving the purity-informativeness dilemma in query selection, which shows that MQ-Net keeps improving the test accuracy even in a later AL round by finding the best balancing between purity and informativeness in its query set. Together with the results in Section 5.2, we confirm that MQ-Net is robust to the two different distributions—'split-dataset' and 'cross-dataset'—of open-set noise.

### D.2    Results with Varying Noise Ratios

Table 7 summarizes the last test accuracy at the final AL round for three cross-datasets with varying levels of open-set noise. Overall, the last test accuracy of MQ-Net is the best in every case, which

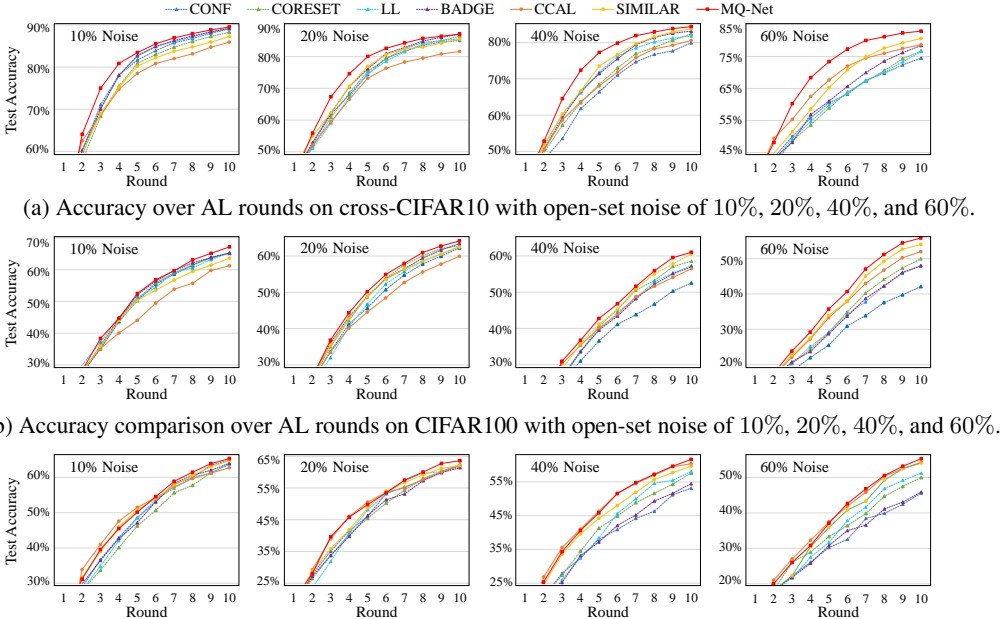

(a) Accuracy over AL rounds on cross-CIFAR10 with open-set noise of 10%, 20%, 40%, and 60%.

(b) Accuracy comparison over AL rounds on CIFAR100 with open-set noise of 10%, 20%, 40%, and 60%.

(c) Accuracy comparison over AL rounds on ImageNet with open-set noise of 10%, 20%, 40%, and 60%.

Figure 5: Test accuracy over AL rounds for the three *cross-datasets*, CIFAR10, CIFAR100, and ImageNet, with varying open-set noise ratios.

Table 7: Last test accuracy (%) at the final round for three cross-datasets: CIFAR10, CIFAR100, and ImageNet50 mixed with the merger of LSUN and Places365. The best results are in bold, and the second best results are underlined.

| Datasets | | Cross-CIFAR10 | | | | Cross-CIFAR100 | | | | Cross-ImageNet50 | | | |
|---|---|---|---|---|---|---|---|---|---|---|---|---|---|
| Noise Ratio | | 10% | 20% | 40% | 60% | 10% | 20% | 40% | 60% | 10% | 20% | 40% | 60% |
| Standard AL | CONF | 89.04 | 85.09 | 79.92 | 74.48 | 65.17 | 62.24 | 52.52 | 42.13 | 64.92 | 61.92 | 53.60 | 45.64 |
| | CORESET | 88.26 | 86.38 | 82.36 | 76.71 | 65.13 | 62.83 | 58.56 | 49.98 | 63.88 | 62.40 | 57.60 | 50.02 |
| | LL | 89.06 | 85.65 | 81.81 | 76.52 | 65.23 | 62.64 | 57.32 | 48.07 | 63.68 | 62.32 | 58.08 | 51.24 |
| | BADGE | 89.2 | 87.07 | 83.14 | 78.38 | 65.27 | 63.42 | 57.01 | 48.07 | 64.04 | 61.40 | 54.48 | 45.92 |
| Open-set AL | CCAL | 85.89 | 81.62 | 80.55 | 78.68 | 61.22 | 59.91 | 56.47 | 52.01 | 62.72 | 62.20 | 60.40 | 54.32 |
| | SIMILAR | 87.24 | 85.50 | 83.80 | 80.58 | 63.61 | 62.46 | 60.52 | 54.05 | 64.72 | 62.04 | 59.68 | 54.05 |
| Proposed | **MQ-Net** | **89.49** | **87.12** | **84.39** | **82.88** | **67.17** | **64.17** | **61.01** | **55.87** | **65.36** | **63.60** | **61.68** | **55.28** |

shows that MQ-Net keeps finding the best trade-off between purity and informativeness in terms of AL accuracy regardless of the noise ratio. The performance improvement becomes larger as the noise ratio increases. Meanwhile, CCAL and SIMILAR are even worse than the four standard AL approaches when noise ratio is less than or equal to 20%. This trend indicates that focusing on informativeness is more beneficial than focusing on purity when the noise ratio is small.

# E    In-depth Analysis of CCAL and SIMILAR in a Low-noise Case

In the low-noise case, the standard AL method, such as CONF, can query many IN examples even without careful consideration of purity. As shown in Table 8, with 10% noise, the ratio of IN examples in the query set reaches 75.24% at the last AL round in CONF. This number is farily similar to 88.46% and 90.24% in CCAL and SIMILAR, respectively. In contrast, with the high-noise case (60% noise), the difference between CONF and CCAL or SIMILAR becomes much larger (*i.e.*, from 16.28% to 41.84% or 67.84%). That is, considering mainly on purity (not informativeness) may not be effective with the low-noise case. Therefore, especially in the low-noise case, the two purity-focused methods, SIMILAR and CCAL, have the potential risk of overly selecting less-informative IN examples that the model already shows high confidence, leading to lower generalization performance than the standard AL methods.

In contrast, MQ-Net outperforms the standard AL baselines by controlling the ratio of IN examples in the query set to be very high at the earlier AL rounds but moderate at the later AL rounds; MQ-Net

Table 8: Test accuracy and ratio of IN examples in a query set for the split-dataset setup on CIFAR10 with open-set noise of 10% and 60%. "%IN in $S_Q$" means the ratio of IN examples in the query set.

| Noise Ratio | Method | Round | 1 | 2 | 3 | 4 | 5 | 6 | 7 | 8 | 9 | 10 |
|---|---|---|---|---|---|---|---|---|---|---|---|---|
| 10% | CONF | Acc | 62.26 | 74.77 | 80.81 | 84.52 | 86.79 | 88.98 | 90.58 | 91.48 | 92.36 | 92.83 |
| | | %IN in $S_Q$ | 87.52 | 82.28 | 80.84 | 79.00 | 75.16 | 76.21 | 74.08 | 74.61 | 74.00 | 75.24 |
| | CCAL | Acc | 61.18 | 71.80 | 78.18 | 82.26 | 84.96 | 86.98 | 88.23 | 89.22 | 89.82 | 90.55 |
| | | %IN in $S_Q$ | 89.04 | 88.48 | 89.12 | 88.64 | 89.52 | 88.80 | 90.44 | 88.08 | 88.64 | 88.46 |
| | SIMILAR | Acc | 63.48 | 73.51 | 77.92 | 81.54 | 84.04 | 86.28 | 87.61 | 88.46 | 89.20 | 89.92 |
| | | %IN in $S_Q$ | 91.44 | 91.04 | 91.52 | 92.56 | 92.61 | 91.40 | 92.24 | 90.64 | 90.75 | 90.24 |
| | MQ-Net | Acc | 61.59 | 73.30 | 80.36 | 84.88 | 87.91 | 90.10 | 91.26 | 92.23 | 92.90 | 93.10 |
| | | %IN in $S_Q$ | 94.76 | 93.28 | 88.84 | 86.96 | 82.04 | 79.60 | 77.24 | 76.92 | 79.00 | 75.80 |
| 60% | CONF | Acc | 56.14 | 65.17 | 69.60 | 73.63 | 76.28 | 80.27 | 81.63 | 83.69 | 84.88 | 85.43 |
| | | %IN in $S_Q$ | 37.44 | 32.20 | 28.16 | 25.40 | 25.64 | 20.08 | 20.88 | 17.00 | 18.04 | 16.28 |
| | CCAL | Acc | 56.54 | 66.97 | 72.16 | 76.32 | 80.21 | 82.94 | 84.64 | 85.68 | 86.58 | 87.49 |
| | | %IN in $S_Q$ | 41.92 | 38.52 | 39.76 | 41.20 | 38.64 | 42.16 | 42.24 | 40.32 | 42.24 | 41.84 |
| | SIMILAR | Acc | 57.60 | 67.58 | 71.95 | 75.70 | 79.67 | 82.20 | 84.17 | 85.86 | 86.81 | 87.58 |
| | | %IN in $S_Q$ | 56.08 | 61.08 | 67.12 | 66.56 | 67.32 | 67.28 | 68.08 | 67.00 | 68.16 | 67.84 |
| | MQ-Net | Acc | 54.87 | 68.49 | 75.84 | 80.16 | 83.37 | 85.64 | 87.56 | 88.43 | 89.26 | 89.51 |
| | | %IN in $S_Q$ | 82.80 | 79.92 | 65.88 | 55.40 | 52.00 | 47.52 | 46.60 | 41.44 | 36.52 | 35.64 |

achieves a higher ratio of IN examples in the query set than CONF at every AL round, but the gap keeps decreasing. Specifically, with 10% noise, the ratio of IN examples in the query set reaches 94.76% at the first AL round in MQ-Net, which is higher than 87.52% in CONF, but it becomes 75.80% at the last AL round, which is very similar to 75.24% in CONF. This observation means that MQ-Net succeeds in maintaining the high purity of the query set and avoiding the risk of overly selecting less-informative IN examples at the later learning stage.

# F   In-depth Analysis of Various Purity Scores

The OSR performance of classifier-dependent OOD detection methods, *e.g.*, ReAct, degrades significantly if the classifier performs poorly [39]. Also, the OSR performance of self-supervised OOD detection methods, *e.g.*, CSI, highly depends on the sufficient amount of clean IN examples [35, 36]. Table 9 shows the OOD detection performance of two OOD detectors, ReAct and CSI, over AL rounds with MQ-Net. Notably, at the earlier AL rounds,

Table 9: OOD detection performance (AUROC) of two different OOD scores over AL rounds with MQ-Net.

| Dataset | | CIFAR10 (4:6 split), 40% Noise | | | | |
|---|---|---|---|---|---|---|
| Round | | 2 | 4 | 6 | 8 | 10 |
| MQ-Net | ReAct | 0.615 | 0.684 | 0.776 | 0.819 | 0.849 |
| | CSI | 0.745 | 0.772 | 0.814 | 0.849 | 0.870 |

CSI is better than ReAct, meaning that self-supervised OOD detection methods are more robust than classifier-dependent methods when the amount of labeled data is small. Thus, the versions of MQ-Net using CSI as the purity score is better than those using ReAct, as shown in Section 5.4.

# G   Additional Experiment Results

## G.1   AL Performance with More Rounds

Figure 6 shows the test accuracy over longer AL rounds for the split-dataset setup on CIFAR10 with an open-set noise ratio of 40%. Owing to the ability to find the best balance between purity and informativeness, MQ-Net achieves the highest accuracy on every AL round. The purity-focused approaches, CCAL and SIMILAR, lose their effectiveness at the later AL rounds, compared to the informativeness-focused approaches, CONF, CORESET, and BADGE; the superiority of CONF, CORESET, and BADGE over CCAL and SIMILAR gets larger as the AL round proceeds, meaning that fewer but highly-informative examples are more beneficial than more but less-informative examples for model generalization as the model performance converges. However, with low (*e.g.*, 20%) open-set noise cases, most OOD examples are selected as a query set and removed from the unlabeled set in a few additional AL rounds, because the number of OOD examples in the unlabeled set is originally small. Thus, the situation quickly becomes similar to the standard AL setting.

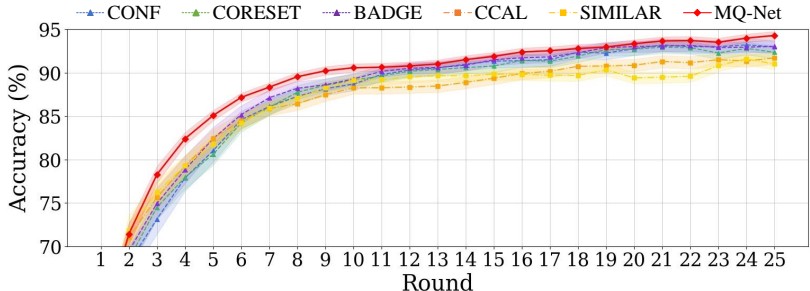

Figure 6: Test accuracy over longer AL rounds for the split-dataset setup on CIFAR10 with an open-set noise ratio of 40%. 500 examples are selected as a query set in each AL round.

## G.2 Standard Deviations of Main Results

Table 10 repeats Table 1 with the addition of the standard deviations. Note that the standard deviations are very small, and the significance of the empirical results is sufficiently high.

## H  Limitation and Potential Negative Societal Impact

**Limitation.** Although MQ-Net outperforms other methods on multiple pairs of noisy datasets under the open-set AL settings, there are some issues that need to be further discussed. First, the performance gap between standard AL without open-set noise and open-set AL still exists. That is, we could not *completely* eliminate the negative effect of open-set noise. Second, although we validated MQ-Net with many OOD datasets, its effectiveness may vary according to the types of the OOD datasets. Formulating the effectiveness of MQ-Net based on the characteristics of a given pair of IN and OOD datasets can be an interesting research direction. Third, we regarded the OOD examples in a query set to be completely useless in training, but recent studies have reported that the OOD examples are helpful for model generalization [49, 50, 51, 15]. Therefore, analyzing how to use OOD examples for model generalization and sample selection in AL can also be an interesting research direction.

**Potential Negative Societal Impact.** As in *all* active learning approaches, since MQ-Net requires human oracles to label each of queried data examples, the oracle can see these examples in a database, even if the proportion of the revealed examples could be very small. Then, if the oracle is not trustworthy, there may be a leak of information. Therefore, in active learning, not specifically confined to MQ-Net, this privacy breach issue should be carefully considered, especially if the database contains private or sensitive information.

Table 10: Last test accuracy (%) at the final round with the standard deviations for CIFAR10, CIFAR100, and ImageNet.

| Datasets | | CIFAR10 (4:6 split) | | | | CIFAR100 (40:60 split) | | | | ImageNet (50:950 split) | | | |
|---|---|---|---|---|---|---|---|---|---|---|---|---|---|
| Noise Ratio | | 10% | 20% | 40% | 60% | 10% | 20% | 40% | 60% | 10% | 20% | 40% | 60% |
| Non-AL | RANDOM | 89.83±0.61 | 89.06±0.59 | 87.73±0.83 | 85.64±0.90 | 60.88±0.67 | 59.69±0.61 | 55.52±0.46 | 47.37±0.80 | 62.72±0.68 | 60.12±0.23 | 54.04±0.56 | 48.24±0.62 |
| Standard AL | CONF | 92.83±0.49 | 91.72±0.55 | 88.69±0.75 | 85.43±0.82 | 62.84±0.52 | 60.20±0.79 | 53.74±0.38 | 45.38±0.64 | 63.56±0.34 | 62.56±0.56 | 51.08±0.63 | 45.04±0.90 |
| | CORESET | 91.76±0.55 | 91.06±0.57 | 89.12±0.72 | 86.50±0.79 | 63.79±0.45 | 62.02±0.54 | 56.21±0.43 | 48.33±0.51 | 63.64±0.60 | 62.24±0.23 | 55.32±0.56 | 49.04±0.42 |
| | LL | 92.09±0.57 | 91.21±0.57 | 89.41±0.62 | 86.95±0.58 | 65.08±0.54 | 64.04±0.48 | 56.27±0.53 | 48.49±0.57 | 63.28±0.23 | 61.56±0.46 | 55.68±0.60 | 47.30±0.72 |
| | BADGE | 92.80±0.49 | 91.73±0.45 | 89.27±0.54 | 86.83±0.38 | 62.54±0.47 | 61.28±0.43 | 55.07±0.52 | 47.60±0.42 | 64.84±0.29 | 61.48±0.29 | 54.04±0.56 | 47.80±0.47 |
| Open-set AL | CCAL | 90.55±0.35 | 89.99±0.45 | 88.87±0.70 | 87.49±0.75 | 61.20±0.39 | 61.16±0.57 | 56.70±0.56 | 50.20±0.49 | 61.68±0.26 | 60.70±0.38 | 56.60±0.85 | 51.16±0.74 |
| | SIMILAR | 89.92±0.55 | 89.19±0.59 | 88.53±0.66 | 87.38±0.73 | 60.07±0.35 | 59.89±0.53 | 56.13±0.41 | 50.61±0.34 | 63.92±0.20 | 61.40±0.36 | 56.48±0.78 | 52.84±0.62 |
| Proposed | MQ-Net | **93.10**±0.40 | **92.10**±0.45 | **91.48**±0.56 | **89.51**±0.58 | **66.44**±0.55 | **64.79**±0.44 | **58.96**±0.36 | **52.82**±0.46 | **65.36**±0.42 | **63.08**±0.23 | **56.95**±0.56 | **54.11**±0.45 |
| % improve over 2nd best | | 0.32 | 0.40 | 2.32 | 2.32 | 2.09 | 1.17 | 3.99 | 4.37 | 0.80 | 1.35 | 0.62 | 2.40 |
| % improve over the least | | 3.53 | 3.26 | 3.33 | 4.78 | 10.6 | 8.18 | 9.71 | 16.39 | 5.97 | 3.92 | 11.49 | 20.14 |