# OpenReview forum: "Meta-Query-Net: Resolving Purity-Informativeness Dilemma in Open-set Active Learning"
_NeurIPS.cc/2022/Conference — NeurIPS 2022 Accept_

### Official Review · Reviewer_wpJ5 · 2022-06-30

**Rating:** 7
**Confidence:** 3
**Soundness:** 3 good
**Presentation:** 3 good
**Contribution:** 3 good

**Summary:**

The paper discusses the purity-informativeness dilemma for open-set active learning, where excessively increasing purity can lead to loss of informativenss. As mitigation they propose a new meta-heuristic, which learns to monotonically combine a purity and an informativeness metric, called MQ-Net. MQ-Net is trained on the acquired query sets alternatingly to utilizing it for query selection.

**Questions:**

Major questions:
* could you additionally provide the standard deviation across the five different runs?
* what computing infrastructure was used? In particular, I wonder about the resource requirements for experiments conducted on the ImageNet dataset
* how do you do the z-score normalization, i.e., over which parts do you compute mean and standard deviation?


Minor questions:
* Figure 4: it appears that all red OOD samples received a high purity score - shouldn't it be vice-versa?
* in Theorem 4.1., the constraint that the activation function needs to be monotonically non-decreasing is not mentioned (it is however in the appendix). This is an important requirement, and while most commonly used activation functions are monotonically non-decreasing, some, e.g., Swish, are not
* in the problem statement, you introduce different costs for annotation of OOD and IN examples. To me it is not clear what would motivate this. In the experiments, you then set same costs for OOD and IN labeling, as in related work.

Minor comments:
* L52-53: One of the "HP" should be something else
* L80: directions -> direction
* L89: k-MEANS -> k-MEANS++
* L115: more higher -> higher
* L120: delimma -> dilemma
* L187: a -> the
* L324-325: this could make a good plot, e.g., x=AL round, y=noise ratio, color=slope
* L363: legit -> legitimate / valid
* Figure 4 is very small and thus hard to read

**Limitations:**

Yes, although the discussion is only present in the appendix.

**Strengths And Weaknesses:**

Strengths
* the paper addresses a practically relevant problem
* the proposed heuristic intuitively makes sense, and is compatible with different choices of purity or informativeness measures
* there is small overhead of the heuristic and it seems to be easy to implement

Weaknesses:
* in contrast to the answers in the checklist, no code is provided. It is also unclear how hyperparameters of the MQ-Net were chosen, and there is no report about the resource usage.
* without reporting standard deviations, it is hard to assess the significance of the empirical results - since the authors already report running the experiments five times with different random seeds it is unclear why they did decide to do so; in addition there is also no comparison against the performance of a random acquisition baseline - having it would justify the necessecity of using an active learning strategy at all
* some parts of the paper could be improved in readability, e.g., in the problem statement, $X_{IN}$ is initially defined as set of in-distribution samples (without their label), but later one, $T_{IN}$ coming from the same distribution $D_{IN}$ contains pairs of (input, target).

---

> ### Author Response · Authors · 2022-08-02
> **Response to Reviewer wpJ5**
>
> We are very glad that you have acknowledged the main contribution of this paper. At the same time, we deeply appreciate your valuable comments and reasonable concerns. During the rebuttal process, we have already addressed all of your concerns in the revised paper. Therefore, we look forward to hearing your positive feedback.
>
> ### Major Concerns (Q1~3).
> `Q1-1. No code is provided and no report about the resource usage.`
>
> Thank you very much for helping us improve our paper. We provide our code at [the link](https://anonymous.4open.science/r/MQNet-43E6/). All methods are implemented with PyTorch 1.8.0 and executed using a single NVIDIA Tesla V100 GPU. The experiments for ImageNet could be run smoothly using this resource. We updated this information about the code and resource usage in the revised paper with the R4Q1 mark.
>
> `Q1-2. It is also unclear how hyperparameters of the MQ-Net were chosen.`
>
> Since MQ-Net is trained on low-dimensional meta-input, we decided to use a shallow MLP architecture with a layer number of 2, a hidden dimension size of 64, and the ReLU activation function. The hyperparameters of MQ-Net including its architecture and the optimization configurations are specified in Appendix C.
>
> `Q2. No standard deviation and random baseline.`
>
> Thanks again for pointing out these issues. We added the error bar and the result of the random baseline in Figure 3 and Table 8 in the revised version. Table 8 in the supplementary material repeats Table 1 with the addition of the standard deviations. Evidently, the standard deviations are very small, and the significance of the empirical results is sufficiently high.
>
> `Q3. How do you do the z-score normalization, i.e., over which parts do you compute mean and standard deviation?`
>
> We conduct z-score normalization for each scalar score $O(x)$ and $Q(x)$. That is, we iteratively compute the mean and standard deviation over the unlabeled examples for every AL round. The mean and standard deviation are computed before the meta-training, and they are used for the z-score normalization at that round. We further clarified this procedure in the revised paper in Lines 237-240 with the R4Q3 mark.
>
>
> ### Minor Questions (Q4~8).
> `Q4. Figure 4: it appears that all red OOD samples received a high purity score - shouldn't it be vice-versa?`
>
> In Figure 4, most red OOD examples received around 0.7~0.8 purity scores which are regarded as being low in CSI-based purity scores. Some OOD examples may receive a high purity score over 0.9, since the open-set recognition performance of CSI is not very accurate in AL due to the lack of a sufficient amount of clean labeled examples.
>
> `Q5. In Theorem 4.1, the constraint that the activation function needs to be monotonically non-decreasing is not mentioned (it is however in the appendix).`
>
> Thank you for pointing out this important issue. According to your suggestion, we fixed Theorem 4.1 by adding the constraint of an activation function. See the updated Theorem 4.1 in the revised paper.
>
> `Q6. In the problem statement, different annotation costs for annotation of IN and OOD examples are introduced. But in the experiments, the costs for OOD and IN labeling are set as the same.`
>
> We agree and appreciate this comment. According to the reviewer’s suggestion, we conducted additional experiments and analyzed the effect of different costs for querying OOD examples in Appendix F in the revised version. Table 7 summarizes the performance change with four different labeling costs (i.e., 0.5, 1, 2, and 4) for the split-dataset setup on CIFAR10 with an open-set ratio of 40%. Overall, MQ-Net consistently outperformed the four baselines regardless of the labeling costs. Meanwhile, CCAL and SIMILAR were more robust to the higher labeling cost than CONF and CORESET. This is because CCAL and SIMILAR, which favor high purity examples, query more in-distribution examples than CONF and CORESET, so they are less affected by labeling costs, especially when cost $\tilde{c}$ is high.
>
> `Q7. Some parts of the paper could be improved in readability.`
>
> Per your suggestion, we removed an unimportant notation, $T_{in}$, in the problem statement.
>
> `Q8. Typos.`
>
> Thank you very much for helping us improve our paper. We fixed all the typos in the revised paper.

---

> > ### Comment · Reviewer_wpJ5 · 2022-08-08
> > **Response**
> >
> > Thank your for your detailed response, as well as the well-signaled updates to your paper! I updated my review scores accordingly.

---

### Official Review · Reviewer_enpC · 2022-07-10

**Rating:** 7
**Confidence:** 2
**Soundness:** 4 excellent
**Presentation:** 3 good
**Contribution:** 3 good

**Summary:**

Active learning typically involves querying and labeling points according to their informativeness, e.g. some notion of uncertainty or diversity. However, when there is out-of-distribution (OOD) data in the unlabeled dataset, such criteria may result in querying these OOD points, which wastes the labeling budget. Recent work has focused on improving the purity of queried points by filtering out all the OOD data and selecting informative in-distribution (IN) points. This paper challenges the notion that filtering OOD data needs to be done to the same extent at each round of active learning. Instead, it may be true that at later rounds, informative but impure OOD data points could be useful to query. There is hence a purity-informativeness dilemma that presents a dynamic tradeoff in the querying process. The paper introduces a meta-model, Meta-Query-Net (MQ-Net) that learns a score that is a function of both purity and informativeness scores updated after each round of AL using the labeled self-validation set already created from previous rounds. They also enforce a skyline constraint to encode ordering of the scores and implement it via nonnegative constraints. Empirically, MQ-Net outperforms both AL that doesn’t account for OOD data, and AL that is OOD aware but static throughout rounds. They also show how MQ-Net gradually prioritizes informativeness more than purity in later rounds, and perform ablation studies with varying purity/informativeness scores, removing the skyline constraint, and replacing the self-validation set with a random validation set.

**Questions:**

**Q1.** Given that MQ-Net requires training a model at each round, how long does MQ-Net take to run versus other baselines like CCAL and SIMILAR?


**Q2.** A running example of what the OOD data and informative versus non-informative data looks like would be helpful. For instance, Figure 1a explains the purity-informative dilemma, but it is not clear what the dataset and task are.


**Q3.** Equation 1 is defined as the optimal query set approach, but is not mentioned otherwise in the paper. It is not clear what the purpose of presenting the upper limit of a query set is. Also, the cost constraint in equation 1 is used in MQ-Net but is not mentioned in that section.

**Q4.** The intuitive interpretation of $L(S_Q)$ was not clear. The main idea from previous sections was that we want a loss that emphasizes informativeness more in later rounds. How does this loss function do that? In particular, it would be good to just have some further discussion about the optimal ranking learned by $L(S_Q)$, case by case. E.g., OOD examples are always scored lower than IN examples, but amongst OOD examples they are roughly ranked by their dot product of purity and information.

**Q5.** Typo: lines 52-53, “The HP-focused approach improves….than the HP-focused one at earlier AL rounds”






**Strengths And Weaknesses:**

**Strengths**

_Significance_: The presence of OOD data can indeed confound standard querying approaches in AL and is an important problem in real datasets. It is unclear how to prioritize potential signal in such data dynamically, which this paper takes a principled step towards.

_Originality_: the idea of using a self-validation set to dynamically meta-score points is novel to me and can inspire new ways of adaptively combining different metrics on unlabeled data.

_Quality_: thorough experiments and ablations convince me of MQ-Net’s utility.

**Weaknesses**

_Quality_: it seems like MQ-Net would be more computationally expensive than previous approaches since a model is trained at each round. It would be interesting to see how much longer MQ-Net takes.

_Presentation_: as someone who does not publish in this area, I had a few comments about clarity (in questions below).

---

> ### Author Response · Authors · 2022-08-02
> **Response to Reviewer enpC**
>
> We deeply appreciate the reviewers’ constructive comments and positive feedback on our manuscript.
>
> `Q1. Computational efficiency of MQ-Net?`
>
> This is a good point. MQ-Net needs one more meta-training phase at every AL round. However, it is not very expensive, because MQ-Net uses a **very light MLP** architecture and the amount of a meta-training set is small. For example, the size of the meta-training set is only 1% of the labeled+unlabeled set for our split CIFAR10/100 experiments.
>
> `Q2. A running example of what the OOD data and informative versus non-informative data looks like would be helpful.`
>
> Thank you very much for your careful comment. As you mentioned, Figure 1(a) is intended to explain the purity-informativeness dilemma, but we couldn’t include the details due to lack of the space. Let us detail our intention to present Figure 1(a). The task is to classify dogs and cats in a given image dataset. (1) The HP-LI subset includes trivial (easy) cases of dogs and cats. (2) The HP-HI subset includes moderate and hard cases of dogs and cats, e.g., properly labeled dog-like cats and cat-like dogs. (3) The LP-HI subset includes other similar animals (e.g., wolves and jaguars) which may share some features with dogs and cats. (4) The LP-LI subset includes other dissimilar animals. Overall, it is clear that HP-HI is the most preferable; however, it is NOT clear which of HP-LI and LP-HI is more preferable. This is defined as the purity-informativeness dilemma. We will add this explanation in the supplementary material or an external web page (e.g., GitHub repository).
>
> `Q3. Equation 1 is defined as the optimal query set approach, but is not mentioned otherwise in the paper. Also, the cost constraint in Equation 1 is used in MQ-Net but is not mentioned in that section.`
>
> Equation 1 formalizes the open-set AL problem. Here, MQ-Net is used to derive a query set $S_Q$ in each round. More specifically, the examples in the order of the meta-score $\Phi(x; w)$ within the budget $b$ form the query set. We expect that this query set is very close to $S_Q^*$ in Equation 1. In Section 4.1, we focused on the training of the MQ-Net itself, and the overall procedure involving the budget was not contained. The overall procedure is clearly described in Appendix B (see the AL procedure pseudocode). We will improve the presentation so that Equation 1 and Section 4.1 can be better connected.
>
>
> `Q4. The intuitive interpretation of L(S_Q) was not clear. The main idea from previous sections was that we want a loss that emphasizes informativeness more in later rounds. How does this loss function do that?`
>
> This is a very good question. $L(S_Q)$ is designed to favor high-loss (i.e., uncertain) examples, which tend to be highly informative, in the meta-model $\Phi(\cdot; w)$. Also, the HP-HI subset is the most preferred by the skyline constraint. Because the backbone classifier is not mature at early rounds, the loss value may not precisely represent the informativeness. As its side effect, simply in-distribution (IN) examples can be selected more often at early rounds than at later rounds. As the classifier becomes mature, the loss value is able to precisely capture the informativeness. Consequently, the informativeness can be properly emphasized at later rounds using Equation (3) **with the varying capability of the classifier**.
>
> Figure 4(a) empirically confirms that informativeness is more emphasized as the training progresses. Also, in Figure 3, the gap between MQ-Net and purity-based methods (CCAL and SIMILAR) becomes larger at later rounds. Another new evidence is provided below. We measure the proportion of IN examples in a query set at each round of MQ-Net. As shown in Table R3, this proportion decreases as the training progresses, because informative rather than pure examples are favored at later rounds.
>
> Table R3: Accuracy and ratio of IN examples in the query set for our split-dataset experiment on CIFAR10 with open-set noise of 40%, where \% $Q_{in}$ means % of IN examples in a query set.
>
> | Method | Round |1|2|3|4|5|6|7|8|9|10|
> |:-----:|:----------:|:--:|:--:|:--:|:--:|:--:|:--:|:--:|:--:|:--:|:--:|
> |MQ-Net |     ACC    |59.6|73.1|79.5|82.9|85.7|88.2|89.3|90.1|90.9|91.5|
> |       | \% $Q_{in}$|88.4|81.1|76.6|72.8|66.2|63.4|57.6|61.4|56.7|57.6|
>
>
>
> `Q5. Typos.`
>
> Thank you very much for helping us improve our paper. The second appearance of HP should be changed to HI, and we fixed the typo.

---

> > ### Comment · Reviewer_enpC · 2022-08-09
> > **Thank you for your response**
> >
> > Thank you so much for your thorough response, which addressed all my concerns. I would like to keep my score and recommend acceptance, given that clarity updates are made (e.g. figure 1 and more explanation of $L(S_Q)$).

---

### Official Review · Reviewer_joSH · 2022-07-10

**Rating:** 5
**Confidence:** 4
**Soundness:** 2 fair
**Presentation:** 3 good
**Contribution:** 2 fair

**Summary:**

This paper introduced the open-set noise problem that might exist in active learning, illustrated the purity-informativeness dilemma and proposed a meta-query (MQ) net as a plugin in normal active learning processes. This paper is well-written before experimental part and easy to follow.

**Questions:**

Most of the questions are listed in previous part.

Questions:

1. The definition of open-set noise (see previous part).

2. The motivation/example of purity-informativeness dilemma contains a contradiction (see pervious part). In my view, in ideal case, it is purity rather than informativeness should be emphasized in latter AL processes.

3. In line 63-64, the author said "The input to the meta-model, which includes the target and OOD labels, is obtained for free from each AL round’s query set by leveraging the multi-round property of AL." This is an advantage, learned from the queried ID and OOD samples. But learning is for updating the META model to better output $\Phi(<P(x), I(x)>;w)$, instead of using it to get better $P(x)$ and $I(x)$, it feels like it is just to train a classifier, but in deep learning tasks, the feature representation and classifier are jointly trained.

4. In line 95, why choose classifier-dependent approach to get a meta-model, what is the motivation? Did the author compared with other density- and self-supervised  based methods?

5. In line 277-278, why use CSI and LL for calculating the purity and informativeness scores? Especially LL, some researches show that LL is not stable due to the joint training with LossNet in some tasks.

6. Is mq-net joint trained with basic classifier (e.g., ResNet18) or not like LL?

7. Equation 3, the situation is both $P(x_i)> P(x_j)$ and $I(x_i) > I(x_j)$, this is apparently to find pareto fronts if one regard it as pareto-optimization problem. Could the author provide some discussions of the situation the number of pareto front set (the data samples that satisfies both $P(x_i)> P(x_j)$ and $I(x_i) > I(x_j)$) less than batch size (labeling budget b in main paper) in active learning process?

8. Is the ResNet18 pre-trained?

9. Did the author conduct repeat trials per experiment? (no error bar problem, mentioned in previous part)

10. The experimental results of the CCAL, SIMILAR and standard AL approaches (mentioned in previous part)

11. In line 284-285, the author already defined the cost of querying ood data samples, why it is not an evaluation metric in later experimental result analysis? It is important if the author could present the cost of querying ood data samples, the less cost means that the labeling budget is less wasted.

Hope the authors could provide convincing responses in rebuttal, I will increase my score if the author could persuade me.

Small typo: Line 261, "OOD" not "ODD".







**Limitations:**

The authors adequately addressed the limitations and potential negative societal impact of their work.

**Strengths And Weaknesses:**

Strengths: The idea of adding a mq net into normal active learning is interesting, it just like LL4AL, which add an extra plugin/network to estimate whether the unlabeled data set belong to ood data or not. The design is both effective and flexible. Additionally, the L(SQ) inherently contains multi-objective optimization (pareto optimization) design ($P(x_i)> P(x_j)$ and $I(x_i) > I(x_j)$ for finding pareto fronts). It is interesting.

Weaknesses: there are still several problems left in this paper.
- Firstly is the definition of the open-set noise problem. I think it is just ID/OOD problems, which is properly defined as class mismatch in CCAL and OOD data scenarios in SIMILAR, there is no need to create a new concept and called it open-set noise problem. This concept is wider, for instance, some instances are really belongs to the ID data distribution but contains noise in x, it is not OOD data but it still contains noise. But the author only conduct experiments follows class mismatch settings in the experimental part.
- Secondly is the motivation of the purity-informativeness dilemma,  1) the author provide an example in Figure 1. This example is not convincing enough since LL4AL and CCAL are both non-typical methods, LL4AL is jointly trained with a LossNet and CCAL use SimCLR/CSI for extracting features, they are not comparable. Additionally, the example only show the first 10 rounds and show low-noise (10% and 20 % ood rate) situations; 2) Why can't we maintain purity all the time and at the same time acquire high informativeness? In addition, if there is a method to achieve the ideal/ Optimal effect, the proportion of OOD sample in unlabeled data pool will naturally be higher and higher, and more attention should be attached to purity.
- Thirdly is the experiments, there is no error bar of the conduct experiments. The experimental results on CCAL and SIMILAR are indeed very strange, on low noise situations (10% and 20% OOD data rate), are even worse than typical uncertainty-based sampling strategy (e.g., CONF), I have contacted the authors of CCAL and SIMILAR to asked them if their models would perform worse than typical uncertainty-based measures like (CONF, ENTROPY), the author of SIMILAR paper said "If there is low-noise then it should only be less challenging and the performance should at least be consistent and better than MARGIN.". CCAL author show me their new experiments on low-noise data scenarios, also better than typical uncertainty-based measures. Since the author didn't provide the code (only provide the pdf version of appendix), I cannot check the implementation. Is it fair comparison?

---

> ### Author Response · Authors · 2022-08-02
> **Response to Reviewer joSH (Part 1)**
>
> We deeply appreciate the reviewers’ valuable comments and reasonable concerns. We hope that they can be resolved through our clarifications in this rebuttal.
>
> ### Major Concerns (Q1~3).
> `Q1. About problem definition: There is no need to create a new concept and call it an open-set noise problem. This concept is wider. For instance, some instances belong to ID but contain noise in x. It is not OOD data but it still contains noise.`
>
> Yes, we deal with the IN/OOD problem, the same setting as in CCAL and SIMILAR. In fact, “open-set noise” is frequently used as a **synonym** of “out-of-distribution (OOD)” data in machine learning literature on open-set recognition [Salehi et al., 2021], OOD detection [Yang et al., 2021], open-set noisy label handling [Wei et al., 2021, Wang et al., 2018], and open-set semi-supervised learning [Saito et al., 2021, Yu et al., 2020, Huang et al., 2021]. In particular, looking at [Saito et al., 2021, Yu et al., 2020, Huang et al., 2021] in which OOD examples are mixed with **clean** IN examples, they use the term “**open-set** semi-supervised learning” just like our paper. When the noise in IN examples is addressed as well, it is common to specify closed-set noise and open-set noise together (e.g., see [Sachdeva et al., 2021]), which is beyond the scope of this paper. Overall, we haven’t created a new wider concept or setting. Following your advice, we will clarify that closed-set noise is not involved in the method.
>
>
> [Salehi et al., 2021] "A unified survey on anomaly, novelty, open-set, and out-of-distribution detection: Solutions and future challenges.", arXiv preprint arXiv:2110.14051, 2021.
> [Yang et al., 2021] "Generalized out-of-distribution detection: A survey," arXiv preprint arXiv:2110.11334, 2021.
> [Wei et al., 2021] "Open-set label noise can improve robustness against inherent label noise," In NeurIPS, 2021.
> [Wang et al., 2018] "Iterative learning with open-set noisy labels," In CVPR, 2018.
> [Saito et al., 2021] "Openmatch: Open-set semi-supervised learning with open-set consistency regularization," In NeurIPS, 2021.
> [Yu et al., 2020] "Multi-task curriculum framework for open-set semi-supervised learning," In ECCV, 2020.
> [Huang et al., 2021] "Trash to Treasure: Harvesting OOD Data with Cross-Modal Matching for Open-Set Semi-Supervised Learning," In ICCV, 2021.
> [Sachdeva et al., 2021] "EvidentialMix: Learning with Combined Open-set and Closed-set Noisy Labels," In WACV, 2021.
>
> `Q2-1. About motivation of the purity-informativeness dilemma: Figure1 is not convincing enough since LL4AL and CCAL are both non-typical methods and they are not comparable. Also, the example only shows the first 10 rounds and shows low-noise (10% and 20 % OOD rate) cases.`
>
> LL is indeed a representative HI-focused method in that it only uses the predicted loss as the informativeness score and there is no purity score combined. CCAL is also a representative HP-focused method since it carefully incorporates the purity score by using CSI. Thus, they are **comparable** from the perspective of showing the dominance between the HI-focused method and the HP-focused method throughout the learning stages (i.e., AL rounds). Our 10-round experiment is a quite typical setting in AL literature [Yoo et al., Moon et al.]. For a higher OOD rate (e.g., 30%), a similar trend was observed, where the cross point appeared at a later round. Following your suggestion, we replaced Figure `1(c)` with the plot for a **30% OOD rate** (see the revised paper).
>
> [Yoo et al.] "Learning loss for active learning.", In CVPR, 2019.
> [Moon et al.] "Confidence-aware learning for deep neural networks.", In ICML, 2020.

---

> > ### Author Response · Authors · 2022-08-02
> > **Response to Reviewer joSH (Part 2)**
> >
> > `Q2-2. About motivation of the purity-informativeness dilemma: Why can’t we maintain purity all the time and at the same time acquire high informativeness? If there is an ideal method, the proportion of OOD samples in the unlabeled data pool will naturally be higher and higher, and more attention should be attached to purity.`
> >
> > Usually, the purity score favors examples for which the model exhibits high confidence (i.e., certain in the model's prediction), while the informativeness score favors examples for which the model exhibits low confidence (i.e., uncertain in the model's prediction). That is, **an opposite trend between the two scores is natural**, e.g.,  if an example shows a high purity score, then its informativeness score is likely to be low. Therefore, it is very difficult to achieve high purity and high informativeness all the time.
> >
> > Favoring high purity over the AL rounds is not challenging because we can select a query set with high purity by including only easy examples in an unlabeled set. However, at the later round, this strategy will not make a significant gain in the model performance due to the low informativeness in the selected set. We empirically observed that, as the model performance increases, ‘fewer but highly-informative’ examples are more impactful than ‘more but less-informative’ examples in terms of improving the model performance. Therefore, it is necessary to emphasize informativeness in later AL rounds even at the risk of choosing OOD examples.
> >
> > For the second question, the size of the unlabeled set is assumed to be considerably larger than those of the query set and the labeled set, e.g., vast amounts of unlabeled images collected by web-crawling. Then, even though there is an ideal method, the proportion of OOD examples in the unlabeled pool would **slightly** increase throughout the AL rounds.
> >
> > `Q3-1. About experiment results: There is no error bar. The author didn't provide the code.`
> >
> > We added the error bar in Figure 3 and the standard deviation in Table 8 (supplementary material) in the revised version. We are sorry to miss our source code and now provide it at [the link](https://anonymous.4open.science/r/MQNet-43E6/) (updated in the revised paper with the R2Q3 mark). For implementing SIMILAR and CCAL, we used the same source code available at their official Github links (SIMILAR: https://github.com/decile-team/distil and CCAL: https://github.com/RUC-DWBI-ML/CCAL).
> >
> > `Q3-2. About experiment results: The experimental results on CCAL and SIMILAR are strange on low noise situations (10% and 20% OOD rate), which are even worse than typical uncertainty-based sampling strategy (e.g., CONF). SIMILAR authors said "If there is low-noise then it should only be less challenging and the performance should at least be consistent and better than MARGIN." CCAL author showed me their new experiments on low-noise data scenarios, also better than typical uncertainty-based measures. Is it a fair comparison?`
> >
> > We appreciate the reviewer for these careful comments and answer your questions in two perspectives.
> >
> > (1) We would like to clarify that a low performance of CCAL is **also reported in their original paper [Du et al., 2021]**. See the left-most plot (20% OOD rate) of Figure 4 for CIFAR-10. The accuracy of CCAL is lower than those of several baselines. For your convenience, here is [the link](https://bit.ly/3cXQ3Cm) to Figure 4. For SIMILAR, a low OOD rate was not considered in their original paper [Kothawade et al., 2021]. We are not aware of the new experiment results which the reviewer received from the CCAL authors, because they are unpublished, private communications.
> >
> > [Kothawade et al. 2021] "Similar: Submodular information measures based active learning in realistic scenarios." In NeurIPS, 2021
> > [Du et al. 2021] "Contrastive coding for active learning under class distribution mismatch.", In ICCV, 2021.

---

> > > ### Author Response · Authors · 2022-08-02
> > > **Response to Reviewer joSH (Part 3)**
> > >
> > > (2) Moreover, at the initial phase of our work, we had thought that a low OOD rate was less challenging, just like the SIMILAR authors thought. However, it turned out that our initial thought was wrong for the following reason.
> > >
> > > In the low-noise case, the standard AL methods such as CONF and MARGIN can query many IN examples **even without careful consideration of purity**. As shown in Table R1, with 10% noise, the ratio of IN examples in the query set reaches 75.2% at the last AL round in CONF. This number is farily similar to 88.4% and 90.2% in CCAL and SIMILAR respectively. In contrast, the difference between CONF and CCAL or SIMILAR becomes much larger (i.e., from 16.2% to 41.8% or 67.8%) with the high-noise case (60% noise in Table R2). That is, considering only purity (but not informativeness) may not be effective with the low-noise case. Therefore, especially in the low-noise case, the two purity-based methods, SIMILAR and CCAL, have the potential risk of **overly selecting less-informative IN examples** that the model already shows high confidence, leading to lower generalization performance than the standard AL methods.
> > >
> > > Overall, putting these two facts together, we believe that the low performance of CCAL and SIMILAR on a low OOD rate is not strange and hope that our analysis is persuasive to you.
> > >
> > > Table R1: Accuracy and ratio of IN examples in the query set for our split-dataset experiment on CIFAR10 with open-set noise of **10%**, where \% $Q_{in}$ means % of IN examples in a query set.
> > >
> > > | Method | Round |1|2|3|4|5|6|7|8|9|10|
> > > |:-----:|:----------:|:--:|:--:|:--:|:--:|:--:|:--:|:--:|:--:|:--:|:--:|
> > > | CONF  |     ACC    |62.3|74.8|80.8|84.5|86.8|89.0|90.6|91.5|92.4|92.8|
> > > |       | \% $Q_{in}$|87.6|82.2|80.8|79.0|75.2|76.2|74.0|74.6|74.0|**75.2**|
> > > | CCAL  |     ACC    |61.2|71.8|78.2|82.3|85.0|87.0|88.2|89.2|89.8|90.6|
> > > |       | \% $Q_{in}$|89.0|88.4|89.2|88.6|89.6|88.8|90.4|88.0|88.6|**88.4**|
> > > |SIMILAR|     ACC    |63.5|73.5|77.9|81.5|84.0|86.3|87.6|88.5|89.2|89.9|
> > > |       | \% $Q_{in}$|91.4|91.0|91.6|92.6|92.6|91.4|92.2|90.6|90.8|**90.2**|
> > >
> > > Table R2: Accuracy and ratio of IN examples in the query set for our split-dataset experiment on CIFAR10 with open-set noise of **60%**, where \% $Q_{in}$ means % of IN examples in a query set.
> > >
> > > | Method | Round |1|2|3|4|5|6|7|8|9|10|
> > > |:-----:|:----------:|:--:|:--:|:--:|:--:|:--:|:--:|:--:|:--:|:--:|:--:|
> > > | CONF  |     ACC    |56.1|65.2|69.6|73.6|76.3|80.3|81.6|83.7|84.9|85.4|
> > > |       | \% $Q_{in}$|37.4|32.2|28.2|25.4|25.6|20.0|20.8|17.0|18.0|**16.2**|
> > > | CCAL  |     ACC    |56.5|67.0|72.2|76.3|80.2|82.9|84.6|85.7|86.6|87.5|
> > > |       | \% $Q_{in}$|42.0|38.6|39.8|41.2|38.6|42.2|42.2|40.4|42.2|**41.8**|
> > > |SIMILAR|     ACC    |57.6|67.6|72.0|75.7|79.7|82.2|84.2|85.9|86.8|87.4|
> > > |       | \% $Q_{in}$|56.0|61.0|67.2|66.6|67.4|67.2|68.0|67.1|68.2|**67.8**|
> > >
> > >
> > >
> > >
> > > ### Minor Questions (Q4~11).
> > > `Q4. In lines 63-64, learning is for updating the META model to better output Φ(<P(x),I(x)>; w), instead of using it to get better P(x) and I(x). It feels like it is just to train a classifier, but in deep learning tasks, the feature representation and classifier are jointly trained.`
> > >
> > > Because $P(x)$ and $I(x)$ are obtained from the classifier, $P(x)$ and $I(x)$ get better as the training progresses (with more labeled examples). Of course, the meta-model $\Phi(\langle P(x),I(x) \rangle; w)$ is improved to produce better prioritization. Thus, the score functions and the meta-model are **improved together**, as you precisely expect.
> > >
> > > `Q5. In line 95, why choose a classifier-dependent approach to get a meta-model, what is the motivation?`
> > >
> > > In fact, we didn’t choose a classifier-dependent approach. Line 95 (93 in the revised version) is just the introduction of OSR methods in the related work section.
> > >
> > > `Q6. In line 277-288, why did you use CSI and LL for purity and informativeness scores, respectively?`
> > >
> > > As shown in Section 5.4, we used CONF and LL as the informativeness scores, and CSI and ReAct as the purity scores. We choose the combination of LL and CSI as the default setting of MQ-Net, since it shows good overall accuracy, as reported in Table 2. The other combinations also showed better accuracy than the baselines.
> > >
> > > `Q7. Is MQ-Net jointly trained with the backbone classifier? or not like LL?`
> > >
> > > MQ-Net is disjointly trained with the backbone classifier. That is, the training procedure alternates between the classifier and MQ-Net, and the details can be found in the algorithm pseudocode in Appendix B.

---

> > > > ### Author Response · Authors · 2022-08-02
> > > > **Response to Reviewer joSH (Part 4)**
> > > >
> > > > `Q8. Equation 3 looks similar to find pareto fronts. Could the author provide some discussions about the situation that the size of a pareto front set is less than the batch size in the active learning process?`
> > > >
> > > > This is a good point. Equation 3 is similar to find pareto fronts, but it is slightly different. The pareto front is a set of examples having at least one dominance in purity or informativeness over all other examples (see [Liu et al., 2015] for details). However, the skyline constraint in Equation 3 is just to ensure the output score of MQ-Net to satisfy $\Phi (z_{x_i})$ $>$ $\Phi(z_{x_j})$ if $P(x_i)>P(x_j)$ **AND** $I(x_i)>I(x_j)$ for all $i$, $j$. That is, the Pareto front does not necessarily have to be the set with the highest MQ-Net score. Thus, regardless of the size of a pareto front set, MQ-Net just queries examples in the order of their meta-scores within the budget (i.e., batch size).
> > > >
> > > > [Liu et al., 2015] "Finding pareto optimal groups: Group-based skyline." In VLDB, 2015.
> > > >
> > > >
> > > > `Q9. Is ResNet18 pre-trained?`
> > > >
> > > > No, we did not use any pre-trained networks.
> > > >
> > > > `Q10. Did the author conduct repeat trials per experiment?`
> > > >
> > > > Of course. We clarified it in Line 294-295. We also added the error bar and standard deviation in the revised version.
> > > >
> > > > `Q11. In line 284-285, the author already defined the cost of querying OOD data samples. Why is it not an evaluation metric in later experimental result analysis?`
> > > >
> > > > Thank you very much for helping us improve our paper. According to the reviewer’s suggestion, we conducted additional experiments and analyzed the effect of different costs for querying OOD examples in Appendix F in the revised version. Table 7 summarizes the performance change with four different labeling costs (i.e., 0.5, 1, 2, and 4) for the split-dataset setup on CIFAR10 with an open-set ratio of 40%. Overall, MQ-Net consistently outperformed the four baselines regardless of the labeling costs. Meanwhile, CCAL and SIMILAR were more robust to the higher labeling cost than CONF and CORESET. This is because CCAL and SIMILAR, which favor high purity examples, query more in-distribution examples than CONF and CORESET, so they are less affected by labeling costs, especially when cost $\tilde{c}$ is high.

---

> > > > > ### Comment · Reviewer_joSH · 2022-08-08
> > > > > **response**
> > > > >
> > > > > I'm glad to see the authors' abundant responses in this rebuttal processes. They answered most of my concerns. But still have several concerns left.
> > > > >
> > > > > Figure 1 changes 20% open-set noise to 30% open-set noise. Also compared with Figure 5. We can see the "cross point" would gradually move backward, in 60% open-set noise, CCAL always perform better than LL. That is, the author build the motivation of HI/HP-dominance on low open-set noise situations, and laterly mentioned that high open-set noise rate is more challenging (in "Response to Reviewer joSH (Part 3) "). This is contradictory. Can author provides further discussions?
> > > > >
> > > > > When I mentioned "LL is not a typical method", refers to its unstable performance, for example, it is heavily influenced by the learning rate (see https://arxiv.org/pdf/2206.12569.pdf as reference, see Figure 5). So I don't think its a good choice to set LL as default setting in your work, since the performance on new tasks cannot be guaranteed.
> > > > >
> > > > > Although 10-rounds setting is used in many research papers, it is still not recommended, since the accuracy-round curves are still far from convergence. There is no guarantee that the performance drop would never occur after 10 rounds.
> > > > >
> > > > > Additionally, here is the newest CCAL: https://ieeexplore.ieee.org/document/9816025.
> > > > >
> > > > > I increase my score to 5.

---

> > > > > > ### Author Response · Authors · 2022-08-09
> > > > > > **Additional Response to Reviewer joSH**
> > > > > >
> > > > > > `Q1.  The author builds the motivation of HI/HP-dominance on low open-set noise situations, and laterly mentioned that high open-set noise rate is more challenging (in "Response to Reviewer joSH (Part 3)"). This is contradictory. Can the author provide further discussions?`
> > > > > >
> > > > > > Thank you very much for your positive feedback and insightful comment. We are very happy to discuss this with you!
> > > > > >
> > > > > > First, the HI/HP-dominance, which we call the purity-informativeness dilemma, occurs **regardless of the noise rate** (i.e., in both low and high noise situations). The reason why we build the motivation on a low noise rate in Figure 1 is just ease of exposition. The HI-focused method, LL, considers only informativeness; that is, LL **entirely** focuses on informativeness. Using LL (or any other similar standard method), it is reasonable to show the dilemma (i.e., a cross point) on a low noise rate, because a HI-**entirely**-focused method hardly beats a HP-focused method at a high noise rate as shown in Figure 5. If we had a HI-**relatively**-focused method, it would be possible to show the dilemma on a high noise rate, too; however, as far as we know, there is no such method.
> > > > > >
> > > > > > Second, regarding our response in Part 3, we intended to say that a low noise rate **could be also challenging** for CCAL and SIMILAR because they tend to choose too many easy IN examples. We did not mention that a high noise rate is more challenging. We conjecture that the comparison between Table R1 and Table R2 (i.e., a larger difference from CONF to CCAL and SIMILAR in Table R2 than in Table R1) is not sufficiently clarified, which intends to confirm that CONF is hard to beat CCAL and SIMILAR on a high noise rate because of an insufficient number of IN examples.
> > > > > >
> > > > > > Overall, we humbly believe that our presentation is **not** contradictory. Of course, we will try our best to clearly deliver our intention. In fact, the page limit is very tight, and the supplementary material is not very accessible. Thus, we plan to add more informal description (including this discussion) in our MQ-Net GitHub repository (not released yet because of double-blind reviewing) and add the link in Introduction of the final version.
> > > > > >
> > > > > >
> > > > > > `Q2. LL is heavily influenced by the learning rate. I don't think it is a good choice to set LL as default setting in your work, since the performance on new tasks cannot be guaranteed.`
> > > > > >
> > > > > > Thank you very much for sharing your concerns on LL. Per your suggestion, we are considering changing the default setting from LL to CONF. Thus, we quickly applied the new (tentative) default setting of MQ-Net to CIFAR10. The resulting plots, which correspond to Figure 3(a), are available at [the link](https://www.dropbox.com/s/uwz3nzo56glpnrz/Figure_overall_accuracy_cifar10_CONF_CSI.pdf?dl=0/). The new default setting looks as good as the current default setting. After all the plots are generated, we will carefully determine which default setting is better for MQ-Net. Overall, we will conduct additional experiments to find out a better default setting and then stick to or change to the better one in the final version.
> > > > > >
> > > > > > `Q3. Although the 10-rounds setting is used in many research papers, it is still not recommended since the accuracy-round curves are still far from convergence. There is no guarantee that the performance drop would never occur after 10 rounds.`
> > > > > >
> > > > > > Thank you very much for your careful comment. We will try to conduct additional experiments with longer AL rounds until the model converges and will include the results in the final version.
> > > > > >
> > > > > > `Q4. Here is the newest CCAL: https://ieeexplore.ieee.org/document/9816025.`
> > > > > >
> > > > > > Thank you very much for letting us know about the new paper. ConAL in the new paper is tested also for semi-supervised learning, which is beyond the scope of our paper. For supervised learning, which is the scope of our paper, the methodology and results of ConAL seem to be the same as those of CCAL. We will definitely cite this paper in the final version.

---

> > > > > > > ### Comment · Reviewer_joSH · 2022-08-09
> > > > > > > **response**
> > > > > > >
> > > > > > > I appreciate the author's response. Propose a framework with flexible settings is good, but most time, if people use your model, they prefer default settings, although LL is good on selected tasks in this paper, there is no gaurantee that they also provide good performance on new tasks. The author can set CONF as default setting and LL as alternative setting. For CCAL, the author can revise the previous citation to the new TPAMI verion, since it is more completed than ICCV version. Looking forward to your revisions!

---

> > > > > > > > ### Author Response · Authors · 2022-08-09
> > > > > > > > **Additional Response to Reviewer joSH**
> > > > > > > >
> > > > > > > > Thank you very much for helping us improve our paper. As you suggested, we will carefully determine which of CONF or LL should become the default or alternative setting, by running another round of experiments. We plan to add a new dataset to double-check the flexibility and generalizability of each setting. Because this set of experiments is expected to take pretty long time, we are not able to deliver a revision within the rebuttal period. However, we will definitely do this job for the camera-ready version. In addition, we already replaced the citation for CCAL with the new TPAMI version. Overall, we again deeply appreciate your time and effort spent on this review!

---

### Official Review · Reviewer_s9UD · 2022-07-12

**Rating:** 6
**Confidence:** 4
**Soundness:** 3 good
**Presentation:** 4 excellent
**Contribution:** 3 good

**Summary:**

This paper proposed the "purity-informativeness dilemma" problem and a meta-learning algorithm - MQ-Net to adaptively balance the purity and informativeness scores for better sample selection. The experimental results verify the reasonability of the concerned problem and the proposed solution.

**Questions:**

As said in the weakness part:

1. Is it necessary to train the MQ-Net in each round? What if we train it in the first round (or first several rounds) and fix it in the remaining round?

2. Figure 4 indicates that the function learned by the MQ-Net is fairly simple. How do the MQ-Net's performances differ from those simple alternatives?


**Limitations:**

The authors mentioned one limitation: they only consider one purity score and one informativeness score as input. They left the multi-score input as future work.

However, the multi-score version is intuitive by adding more input dimensions to MQ-Net. Since the authors had already computed many scores in  Section 5.4, the multi-score version is very convenient to implement. Furthermore, when more scores are used, there would be a score selection problem in MQ-Net. The solution to that problem will increase the quality of the paper.

**Strengths And Weaknesses:**

# Strengths

1. The "purity-informativeness dilemma" problem is essential, and the analysis is convincing.
2. The design and implementation of MQ-Net is reasonable.
3. The active learning architecture is clean and clear, with significantly better performance.

# Weakness

1. The paper does not discuss the benefit of training the MQ-Net in each round.
2. The performance comparison between MQ-Net and other simple alternatives is not discussed. Those alternatives may include  heuristic rules like $\frac{1}{(P(x) - 1) ^2 + (I(x) - 1)^2}$, logistic regressions, and ranking SVM.
3. The architecture of MQ-Net is not clearly stated. The activation functions are not reported, and the layer number is only reported in the appendix.

---

> ### Author Response · Authors · 2022-08-02
> **Response to Reviewer s9UD**
>
> We deeply appreciate the reviewers’ valuable comments and some concerns. We hope that the concerns can be resolved through our clarifications in this rebuttal.
>
> `Q1.Is it necessary to train the MQ-Net in each round? What if we train it in the first round (or first several rounds) and fix it in the remaining round?`
>
> The goal of MQ-Net is to **adaptively** find the best trade-off between purity and informativeness throughout the entire AL period, since the optimal balance varies with respect to the learning stage of the target classifier. Thus, if we fix the MQ-Net after the first round (or first several rounds), it no longer adjusts this trade-off, leading to a suboptimal result. For example, if the MQ-Net emphasizes purity over informativeness at the first round and keeps sticking to the policy, many informative examples would be ignored in query selection at later AL rounds.
>
>
> `Q2. The performance comparison between MQ-Net and other simple alternatives is not discussed. How do the MQ-Net's performances differ from those simple alternatives?`
>
> This is a very good point. As shown in Figure 4, the best trade-off between purity and informativeness differs by the learning stage and the open-set noise ratio. However, if we use heuristic rules as simple alternatives, we should search for the best rules every time we learn a new classifier on new datasets. However, as shown in Figure 3, MQ-Net successfully finds the best trade-off throughout the learning stage with varying OOD ratios by leveraging our meta-learning framework. This **flexibility of MQ-Net** is a clear advantage over the simple alternatives.
>
> `Q3. The architecture of MQ-Net is not clearly stated. The activation functions are not reported, and the layer number is only reported in the appendix.`
>
> We appreciate the reviewer for pointing out this important issue. We used a shallow MLP architecture with a layer number of 2, a hidden dimension size of 64, and the ReLU activation function. We clarified these details of the architecture in Section 5.1 with the R1Q3 mark.
>
> `Q4. The multi-score version is intuitive by adding more input dimensions to MQ-Net. Furthermore, when more scores are used, there would be a score selection problem in MQ-Net. The solution to that problem will increase the quality of the paper.`
>
> Thank you very much for helping us improve our paper. Though using multiple (more than two) scores is a very interesting topic, it seems to be beyond the scope of this paper. We leave this topic for potential future work.

---

### Meta-Review · Area_Chair_x2Ny · 2022-08-26

**Recommendation:** Accept
**Confidence:** Less certain

**Metareview:**

The reviewer agree that the paper studies an important problem, and they supported acceptance for highlighting the trade-off between purity and informativeness, which is both novel and significant.

Among the weaknesses that have been raised, there is
- simpler alternatives to using a their MQ-Net would be a plus (Rev s9UD)
- motivating example is a bit handwavy and no clear theoretical motivation (Rev joSH)
- overall, the improvement with respect to baselines is small [answered in the updated appendix with stdev]
- Theorem 4.1 is fairly straightforward, and monotonic networks have been studied for long

While the reviewers agree that the strength outweight the weaknesses, the authors are invited to take these remarks into account when preparing the final version of their manuscript.

**Award:**

No

---

### Decision · Program_Chairs · 2022-09-14

Accept